

**Disturbance legacies have a stronger effect on future carbon exchange than**
**climate in a temperate forest landscape**
**Running head**: "Disturbance legacies determine C exchange"
Dominik Thom[* 1,2], Werner Rammer[1], Rita Garstenauer[3], Rupert Seidl[1]
[1] Institute of Silviculture, Department of Forest- and Soil Sciences, University of Natural
Resources and Life Sciences (BOKU) Vienna, Peter-Jordan-Straße 82, 1190 Vienna, Austria
[2] Rubenstein School of Environment and Natural Resources, University of Vermont, 308i Aiken
Center, Burlington, VT 05405, USA. Tel: +1 802 557 8221. Fax: +1 802 656 2623. Email:
dominik.thom@uvm.edu
[3] Institute of Social Ecology, Alpen-Adria Universität, 1070 Vienna, Austria
[*] Corresponding author



## Abstract

Forest ecosystems play an important role in the global climate system, and are thus intensively discussed in the context of climate change mitigation. Over the past decades temperate forests were a carbon (C) sink to the atmosphere. However, it remains unclear to which degree this C uptake is driven by a recovery from past disturbances vs. ongoing climate warming, inducing high uncertainty regarding the future temperate forest C sink. Here our objectives were (i) to investigate legacies within the natural disturbance regime by empirically analyzing two disturbance episodes affecting the same landscape 90 years apart, and (ii) to unravel the effects of past disturbances and future climate on $21^{st}$ century forest C uptake by means of simulation modelling. We collected historical data from archives to reconstruct vegetation and disturbance history of a forest landscape in the Austrian Alps from 1905 to 2013. The effect of past legacies and future climate was determined by simulating 32 different combinations of past disturbances (including natural disturbances and management) and future climate scenarios. We found only moderate spatial overlap between two episodes of wind and bark beetle disturbance affecting the landscape in the early $20^{th}$ and $21^{st}$ century, respectively. The future forest C sink was driven by past disturbances, while climate change reduced forest C uptake. Historic management (and its cessation) had a considerably stronger influence on the future C balance than the natural disturbance episodes of the past. We conclude that neglecting disturbance legacies can substantially bias assessments of future forest dynamics.

**Key words**: bark beetles, climate change, forest history, forest management, Kalkalpen National Park, legacy effects, net ecosystem exchange, wind



38 **Copyright statement**

## 1. Introduction

Carbon dioxide ($CO_2$) is responsible for 76% of the global greenhouse gas emissions, and is
thus the single most important driver of anthropogenic climate change (IPCC 2014). Forest
ecosystems take up large quantities of $CO_2$ from the atmosphere, and play a key role in
mitigating climate change (IPCC 2007). During the period 1990 – 2007, established and
regrowing forests were estimated to have taken up 60% of the cumulative fossil carbon
emissions (Pan et al., 2011). This carbon (C) sink strength of forests has further increased in
recent years (Keenan and others 2016). Yet, it is likely that a combination of factors play a role
in the increasing carbon sequestration of forest ecosystems. On the one hand, possible factors
contributing to an increasing sink strength of the biosphere are $CO_2$ (Drake et al., 2011) and
nitrogen (Perring et al., 2008) fertilization, in combination with extended vegetation periods
resulting from climate warming (Keenan et al., 2014). On the other hand, the accelerated carbon
uptake by forests might be a transient recovery effect of past carbon losses from land-use and
disturbances (Erb, 2004; Loudermilk et al., 2013).
For the future, dynamic Global Vegetation Models (DGVMs) frequently suggest a persistent
forest carbon sink (Keenan et al., 2016; Sitch et al., 2008). However, while DGVMs are suitable
for tracking the direct effects of global change, they frequently neglect the effects of
disturbances and their long-term legacy. Both natural and anthropogenic disturbances have
decreased the amount of carbon currently stored in forest ecosystems (Erb et al., 2018; Goetz





et al., 2012; Harmon et al., 1990; Seidl et al., 2014a). The legacy effects of past disturbances
have the potential to significantly influence forest dynamics and alter the trajectories of carbon
uptake in forest ecosystems over time frames of decades and centuries (Gough et al., 2007;
Landry et al., 2016; Seidl et al., 2014b). This is of particular importance for the forests of
Central Europe, which have been markedly affected by anthropogenic (i.e., forest management)
and natural (e.g., wind storms and bark beetles) disturbances over the past centuries (Naudts et
al., 2016; Svoboda et al., 2012). The importance of an improved understanding of past
disturbance dynamics and its impacts on the future carbon cycle is further underlined by the
expectation that climate change will amplify natural disturbance regimes in the future (Seidl et
al., 2017). In this context the role of temporal autocorrelation within disturbance regimes is of
particular relevance, i.e., the influence that past disturbances have on future disturbances at a
given site. Are past disturbances increasing or decreasing the propensity and severity for future
disturbances? And are such temporal autocorrelations influencing the future potential of forests
to take up carbon? The propensity and effect of disturbance interactions across decades remain
understudied to date, largely because of a lack of long-term data on past natural and human
disturbances.
Here we investigate the effect of long-term disturbance legacies on forest ecosystem dynamics,
in order to better understand the drivers of future forest carbon uptake, and thus aid the
development of effective climate change mitigation strategies. In particular, our first objective
was to empirically investigate the temporal interaction of two major episodes of natural
disturbance affecting the same Central European forest landscape 90 years apart (i.e., 1917 –
1923 and 2007 – 2013). We hypothesized a temporal autocorrelation of the two major
disturbance episodes, and specifically an amplifying effect from the earlier disturbance episode
on the later disturbance episode, based on recent observations of centennial disturbance waves
in Europe's forests (Schurman et al., 2018). Our second goal was to quantify the contribution



of past disturbances (both natural and anthropogenic) on the future C uptake of the landscape
under a number of climate change scenarios using simulation modelling. We were particularly
interested in the relative effects of past disturbances and future climate scenarios on the future
forest C sink strength. To that end we reconstructed the vegetation and disturbance history of
the landscape from 1905 to 2013 using historical sources and remote sensing. We subsequently
determined the effect of past disturbances on 21$^{st}$ century C dynamics by simulating forests
from the early 20$^{th}$ century to the end of the 21$^{st}$ century, experimentally altering past
disturbance regimes in a factorial simulation experiment. These analyses were run under
multiple climate scenarios for the 21$^{st}$ century, and focused on Net Ecosystem Exchange (NEE)
(i.e., the net C exchange of the ecosystem with the atmosphere) as the response variable. We
hypothesized that the legacy of past disturbances (management + natural causes) is of
paramount importance for the future carbon sink (Thom et al., 2017a), expecting a saturation
of carbon uptake as the landscape recovers from past disturbances (i.e., a negative but
decreasing NEE through the 21$^{st}$ century). Moreover, we hypothesized a negative impact of
future climate change on carbon uptake as a result of less favorable conditions for carbon-rich
spruce dominated forests (Thom et al., 2017a).

## 103   2. Materials and Methods

### 104   2.1 Study area

We selected a 7,609 ha forest landscape located in the northern front range of the Alps as our
study area (Fig. 1). Focusing on the landscape scale allowed us to mechanistically capture
changes in forest structure and C stocks by jointly considering processes at the large scale such
as disturbances as well as fine scale processes such as competition between individual trees.
The focal landscape is particularly suited to address our research questions as it (i) was affected



by two major episodes of natural disturbance (driven by wind and bark beetles) in the past
century, and (ii) has a varied management history, with intensive management up until 1997,
and then becoming a part of Kalkalpen National Park (KANP), the largest contiguous protected
forest area in Austria. The steep elevational gradient of the study landscape, ranging from 414
m to 1637 m a.s.l., results in large variation in environmental conditions. For instance,
temperatures range from $4.3 – 9.0°C$ and mean annual precipitation sums vary between 1179 –
1648 mm on the landscape. Shallow Lithic and Renzic Leptosols as well as Chromic Cambisols
over calcareous bedrock are the prevailing soil types (Kobler 2004). The most prominent natural
forest types on the landscape are European beech (*Fagus sylvatica* [L.]) dominated forests at
low elevations, mixtures of Norway spruce (*Picea abies* [K.]), silver fir (*Abies alba* [Mill.]) and
European beech at mid-elevations, and Norway spruce dominated forests at high elevations.
These forest types are among the most common ones in Europe, and are highly valuable to
society also from a socio-economic perspective (Hanewinkel et al., 2012).

## 124     2.2 Simulation model

We employed the individual-based forest landscape and disturbance model (iLand) to simulate
past and future forest dynamics at our study landscape. iLand is a high-resolution process-based
forest model, designed to simulate the dynamic feedbacks between vegetation, climate and
disturbance regimes (Seidl et al., 2012a, 2012b). It simulates processes in a hierarchical multi-
scale framework, i.e., considering processes at the individual tree (e.g., growth, mortality as
well as competition for light, water, and nutrients), stand (e.g., water and nutrient availability),
and landscape (e.g., seed dispersal, disturbances) scale as well as their cross-scale interactions.
Competition for resources among individual trees is based on ecological field theory (Wu et al.,
1985). Resource utilization is modelled employing a light use efficiency approach (Landsberg



and Waring, 1997), incorporating the effects of temperature, solar radiation, vapor pressure
deficit, soil water and nutrient availability on a daily basis. Resource use efficiency is further
modified by variation in the atmospheric $CO_2$ concentration. Seeds are dispersed via species-
specific dispersal kernels ($20 \times 20$ m horizontal resolution) around individual mature trees. The
establishment success of the regeneration is constrained by environmental filters (e.g.,
temperature and light availability).
Mortality of trees is driven by stress-induced carbon starvation and also considers a stochastic
probability of tree death depending on life-history traits. Additionally, iLand includes three
submodules to simulate natural disturbances, including wind (Seidl et al., 2014c), bark beetles
(Seidl and Rammer 2017), and wildfire (Seidl et al., 2014b). As wind and bark beetles are of
paramount importance for the past and future disturbance regimes of Central Europe's forests
(Seidl et al., 2014a; Thom et al., 2013), we employed only these two process-based disturbance
submodules in our simulations. The impact of wind disturbance in iLand depends on species-
and size-specific susceptibility (e.g., critical wind speeds of uprooting and stem breakage),
vertical forest structure (e.g., gaps), and storm characteristics (e.g., maximum wind speeds).
The bark beetle module simulates the impact of *Ips typographus* (L.) on Norway spruce, and
thus addresses the effects of the most important bark beetle species in Europe with respect to
area affected and timber volume disturbed (Kautz et al., 2017; Seidl et al., 2009). The model
*inter alia* accounts for insect abundance, phenology and development, as well as emergence
and dispersal. It computes the number of beetle generations and sister broods developed per
year based on the prevailing climate, and considers individual tree defense capacity and
susceptibility. Interactions between wind and bark beetles arise from a high infestation
probability and low defense capacity of freshly downed trees after wind disturbance, while
newly formed gaps (e.g., by bark beetles) increase the exposure of surrounding forests to storm
events.



In addition to the submodules of natural disturbance we used the agent-based forest
management module (ABE) in iLand (Rammer and Seidl, 2015) to simulate past forest
disturbances  by management. ABE enables the dynamic application of generalized stand
treatment programs, including planting, tending, thinning, and harvesting activities. The
dynamically simulated management agent observes constraints at the stand and landscape
scales, such as maximum clearing sizes and sustainable harvest levels. Besides silvicultural
treatments, we used ABE to emulate the past management practice of salvage logging after bark
beetle outbreaks. A detailed description of the implementation of historic management
activities in the simulations can be found in the Supplementary Material (S4).
iLand simulates a closed carbon cycle, tracking C in both aboveground (stem, branch, foliage,
tree regeneration) and belowground live tree compartments (coarse and fine roots).
Decomposition rates of detrital pools are modified by temperature and humidity to allow for
the simulation of C dynamics under changing climatic conditions. Detrital pools include litter
(i.e., dead material from both leaf and fine root turnover) and soil organic matter (Kätterer and
Andrén, 2001) as well as snags and downed coarse woody debris.
iLand has been extensively evaluated against independent data from forest ecosystems of the
northern front range of the Alps using a pattern-oriented modeling approach (Grimm, 2005).
The patterns for which simulations were compared against independent observations include
tree productivity gradients and natural vegetation dynamics (Thom et al., 2017b), wind and bark
beetle disturbance levels and distributions (Seidl and Rammer 2017), as well as management
trajectories (Albrich et al., 2018). A comprehensive documentation of iLand can be found
online at http://iLand.boku.ac.at, where also the model executable and source code are freely
available under a GNU GPL open source license.





## 2.3 Reconstructing forest management and disturbance history

The study area has a long history of intensive timber harvesting for charcoal production, mainly driven by a local pre-industrial iron-producing syndicate. This syndicate was active until 1889, when the land was purchased by the k.k. ("kaiserlich und königlich") Ministry for Agriculture. During the 20[th] century, the majority of the landscape was managed by the Austrian Federal Forests, and only limited areas within the landscape were still under the ownership of industrial private companies (Weichenberger, 1994, 1995; Weinfurter, 2005). Forest management in the late 19[th] and early 20[th] century was strongly influenced by the emerging industrialization. The substitution of wood by mineral coal for heating, but especially for industrial energy supply, changed the focus of forest management from fuel wood to timber production. At the same time, an increase in agricultural productivity (also triggered by input of fossil resources as well as artificial fertilizer) allowed for the abandonment of less productive agricultural plots, often followed by afforestation or natural regrowth of forest vegetation. Consequently, growing stocks increased in many parts of Europe throughout the 20[th] century as the result of increases in both forest extent and density (Bebi et al., 2017). In our study system, the shifting focus from fuel wood to timber production around 1900 was accompanied by the introduction of systematic stand delineation for spatial management planning (Fig. S2) and decadal inventories and forest plan revisions. These documents are preserved in the archives of the Austrian Federal Forests, and were used here to reconstruct past forest vegetation as well as management and disturbance history (see S1, Fig. S2 and S3 in the Supplementary Material for details).

The oldest historic vegetation data available for the landscape were from an inventory conducted between the years 1898 and 1911 and comprised growing stock and age classes for 11 tree species at the level of stand compartments for the entire landscape; we subsequently used the year 1905 (representing the area-weighted mean year of this initial inventory) as the temporal starting point for our analyses (Fig. 2). A major challenge for managers was to extract



resources from remote and inaccessible parts of the topographically highly complex landscape.
The most important means of timber transportation was drifting (i.e., flushing logs down creeks
and streams after artificially damming them). However, this transportation technique was not
feasible for heavy hardwood timber such as beech (Grabner et al., 2004). Consequently,
managers harvested trees selectively, and mainly focused on accessible areas (i.e., stands close
to streams), leading to a bimodal age distribution on the landscape in 1905 with many young
and several old stands (Fig. S8).
In addition to deriving the state of the forest in 1905, we reconstructed management activities
(thinnings, final harvests, artificial regeneration) and natural disturbances (wind and bark
beetles) until 2013. From 1905 to 1917 timber extraction was fairly low. Between 1917 and
1923, however, a major disturbance episode by wind and bark beetles hit the region. Resulting
from a lack of labor force (military draft, malnutrition) in the last year of World War I a major
windthrow in 1917 could not be cleared, and the resulting bark beetle outbreak affected large
parts of the landscape. Overall, wind and bark beetles disturbed approximately one million
cubic meters of timber in our study area between 1917 and 1923 (calculation from archival
sources; Soyka, 1936; Weichenberger, 1994). Consequently, a railroad was installed to access
and salvage the disturbed timber. After the containment of the disturbance in 1923 forest
management resumed at low intensity and no major natural disturbances were recorded.
Following World War II, a network of forest roads was built in order to gradually replace
transportation by railroads. The introduction of motorized chain saws (Fig. 2) further
contributed to an intensification of harvests. By 1971, forest railroads were completely replaced
by motorized transportation on forest roads, resulting in a further increase in the timber
extracted from the landscape (Fig. S9). Timber removals from management as well as natural
disturbances from wind and bark beetles between 1905 and 1997 were reconstructed from
yearly management reviews available from archival sources. With the landscape becoming part



of KANP forest management ceased in 1997. A second major episode of natural disturbances
affected the landscape from 2007-2013, when a large bark beetle outbreak followed three storm
events in 2007 and 2008. This second disturbance episode was reconstructed from disturbance
records of KANP in combination with remote sensing data (Seidl and Rammer, 2016; Thom et
al., 2017b).

**2.4 Landscape initialization and drivers**
The vegetation data for the year 1905 were derived from historical records for 2079 stands with
a median stand area of 1.7 ha. On average over the landscape, the growing stock was 212.3 m³
ha$^{-1}$ in 1905. The most common species were Norway spruce (with a growing stock of on
average 116.3 m³ ha$^{-1}$), European beech (68.0 m³ ha$^{-1}$), and European larch (*Larix decidua*
[Mill.], 21.5 m³ ha$^{-1}$). With an average growing stock of 4.2 m³ ha$^{-1}$ silver fir was considerably
underrepresented on the landscape relative to the potential natural vegetation composition,
resulting from historic clear-cut management and high browsing pressure from deer (see also
Kučeravá and others 2012). Despite these detailed data on past vegetation not all information
for initializing iLand were available from archival sources, e.g., diameters at breast height (dbh)
and height of individual trees, as well as tree positions, regeneration and belowground carbon-
pools had to be reconstructed by other means. To that end we developed a new method for
initializing vegetation in iLand, combining spin-up simulations with empirical reference data
on vegetation state, henceforth referred to as legacy spin-up.
Commonly, spin-ups run models for a certain amount of time or until specified stopping criteria
are reached (e.g., steady-state conditions). The actual model-based analysis is then started from
the thus spun-up vegetation condition (Thornton and Rosenbloom, 2005). This has the
advantage that the model-internal dynamics (e.g., the relationships between the different C and



N pools in an ecosystem) are consistent when the focal analysis starts. However, the thus
derived initial vegetation condition does frequently not correspond well with the vegetation
state observed at a given point in time, and does not account for the legacies of past management
and disturbance. The legacy spin-up approach developed here aims to reconstruct a (partially)
known reference state of the vegetation (e.g., the species composition, age, and growing stock
reconstructed from archival sources for the current analysis) from simulations (Fig. S5). To this
end iLand simulates long-term forest development for each stand, employing an approximation
of the past management and disturbance regime. During the simulations, the emerging forest
trajectory is periodically compared to the respective reference values, and the assumed past
management is adapted iteratively in order to decrease the difference between simulated
vegetation states and reference values. This procedure is executed in parallel for all stands on
the landscape over a long period of time (here: 1000 years), and the simulated vegetation states
best corresponding to the reference values are stored (including individual tree properties,
regeneration, and carbon pools), and later used as initial values for model-based scenario
analyses. A detailed description of the legacy spin-up approach is given in the Supplementary
Material S4.
In simulating 20[th] century forest dynamics we accounted for the abandonment of cattle grazing
and litter raking in forests (Glatzel, 1991) as well as an increasing deposition of nitrogen from
the atmosphere (Dirnböck et al., 2014; Roth et al., 2015). Specifically, we dynamically
modified the annual plant available nitrogen in our simulations based on data of nitrogen
deposition in Austria between 1880 and 2010, with nitrogen input culminating in the mid 1980s,
followed by a decrease and a stabilization after 2000 (Dirnböck et al., 2017). Besides edaphic
factors also an increase in temperature has led to more favorable conditions of tree growth
(Pretzsch et al., 2014). Detailed observations of climate for our study region reach back to 1950
(Thom et al., 2017b), requiring an extension of the climate time series to 1905. We extracted



data from the nearest weather station covering the period from 1905 to present (i.e., Admont,
located approximately 20 km south of our study area), and used its temperature and
precipitation record to sample years with corresponding conditions from the observational
record for our study landscape.
Simulations were run from 1905 until 2099, considering four different climate scenarios for the
period 2013 – 2099. Climate change was represented by three combinations of global
circulation models (GCM) and regional climate models (RCM) under A1B forcing, including
CNRM-RM4.5 (Radu et al., 2008) driven by the GCM ARPEGE, and MPI-REMO (Jacob,
2001), as well as ICTP-RegCM3 (Pal et al., 2007), both driven by the GCM ECHAM5. The
A1B scenario family assumes rapid economic growth with a global population peaking in mid-
century and declining thereafter, and a balanced mix of energy sources being used (IPCC 2000).
With average temperature increases of between +3.1°C and +3.3°C and changing annual
precipitation sums of -87.0 mm to +135.6 mm by the end of the 21$^{st}$ century, the scenarios
studied here are comparable to the changes expected under the representative concentration
pathways RCP4.5 and RCP6.0 for our study region (Thom et al., 2017c). In addition to the three
scenarios of climate change a historic climate scenario was simulated. The years 1950 – 2010
were used to represent this climatic baseline, and were randomly resampled to derive a
stationary climate time series until 2099.

## 2.5 Analyses

To address our first objective and investigate the spatio-temporal interactions of natural
disturbances we used the stand-level records of the two historic disturbance episodes (1917 –
1923 and 2007 – 2013). First, we discretized the information (disturbed/ undisturbed) and
rasterized the stand polygon data to a grid of 10 × 10 m. Subsequently, we used this grid to



calculate an odds ratio for the probability that the two disturbance events affected the same
locations on the landscape (i.e., the odds that areas disturbed in the first episode were disturbed
again in the second episode). We calculated the 95% confidence interval of the odds ratio using
the vcd package in R (Meyer et al., 2016).
To address our second objective and evaluate the impact of past disturbances and future climate
on the 21$^{st}$ century carbon sink strength, we ran simulations under a combination of different
disturbance histories and climate futures. Specifically, we experimentally permutated
disturbances between 1905 and 2013, and analyzed the effect of these permutations by
continuing the simulations until the end of the 21$^{st}$ century. At three points in time a bifurcation
of the disturbance history was considered in the simulation, resulting in eight different pathways
of past landscape dynamics. The three bifurcations were (i) the inclusion or omission of the first
episode of natural disturbance (1917-1923), (ii) a continuation of management until the
founding of the national park 1997 or a cessation of forest management after 1923, and (iii) the
inclusion or omission of the second natural disturbance episode (2007-2013) (Fig. 3). This
factorial permutation of elements of the actual disturbance history of the landscape was chosen
to assess the effects of both past and recent episodes of natural disturbance on future C uptake,
as well as to quantify the role of past management, while accounting for the dynamic
interactions between these factors in the simulation (e.g., between first and second episode of
natural disturbance). After 2013 four different climate scenarios were simulated for all
alternative disturbance histories, to assess the impacts of climate change on the future NEE of
the landscape.
All simulations were started from the landscape conditions in 1905, determined by means of
the legacy spin-up procedure described above. From 1905 to 1923 management and natural
disturbances were implemented in the simulation as recorded in the stand-level archival
sources. After 1923, natural disturbances were simulated dynamically using the respective



iLand disturbance modules. For the second disturbance episode (2007 – 2013) the observed
peak wind speeds for the storms Kyrill (2007), Emma (2008) and Paula (2008) were used in the
simulation (see Seidl and Rammer 2017 for details). Beyond 2013, natural disturbances were
dynamically simulated with iLand. We randomly sampled annual peak wind speeds from the
distribution of years 1924 – 2006 and simulated the wind and bark beetle dynamics emerging
on the landscape (see also Thom et al., 2017a).
Management interventions from 1923 to 1997 were simulated using ABE. The individual
silvicultural decisions where thus implemented dynamically by the management agent in the
model, based on generic stand treatment programs of past management in Austria's federal
forests and the emerging state of the forest. The advantage of this approach was that
management was realistically adapted to different forest states in the simulations, e.g., with
harvesting patterns differing in the runs in which the disturbance episode 1917 – 1923 was
omitted. Moreover, in line with the technical revolutions of the 20$^{th}$ century (Fig. 2) the
simulated management agent was set to account for an intensification of forest management
over time (e.g., a higher number of thinnings and shorter rotation periods). In summary, our
simulation design consisted of 32 combinations of different disturbance histories and climate
futures, which were replicated 20 times (i.e., in total 640 simulation runs) for the years 1905 –
2099 (195 years).
We evaluated the ability of iLand to reproduce past human and natural disturbances as well as
the resultant forest vegetation dynamics on the landscape by comparing simulations of the
baseline scenario (i.e., including historic climate, as well as reconstructed natural disturbances
and forest management) with independent empirical data for different time periods: The
simulated amount of timber extracted was compared to historical records for three time periods
divided by major technical revolutions during the 20$^{th}$ century (Fig. 2). Simulated impacts of
the second disturbance episode (2007 – 2013) on growing stock were compared against



empirical records from KANP. Simulated species shares and total growing stock were
compared against independent data for the year 1905, testing the ability of the legacy spin-up
to recreate the initial vegetation state. Furthermore, simulated species shares and growing stocks
were also related to observations for 1999, i.e., testing the capacity of iLand to faithfully
reproduce forest conditions after 95 years of vegetation dynamics. The results of all these tests
can be found in the Supplement of this study.
We used simulation outputs to investigate the changes in NEE over time and to compare the
different scenarios. NEE denotes the net C flux from the ecosystem to the atmosphere, with
negative values indicating ecosystem C gain (Chapin et al., 2006). To determine the impact of
past disturbances and future climate on the 21$^{st}$ century carbon balance of the landscape, we
first computed the cumulative NEE over the period 2014 – 2099 for each simulation. Next, the
effects of past disturbances and future climate were calculated from mean differences between
the different factor combinations of the simulation experiment with regard to their cumulative
NEE in 2099. P-values were computed by means of permutation-based independence tests
using the coin package (Hothorn and others 2017), and subsequently transformed into
confidence intervals for visualization (Altman 2011). All analyses were performed using the R
language and environment for statistical computing (R Development Core Team 2017).

## 3. Results

### 3.1 Reconstructing historic landscape dynamics

Using iLand, we were able to successfully reproduce historic vegetation and disturbance
dynamics on the landscape. The results from the legacy spin-up revealed a good match with the
species composition and growing stock expected from the historic records for the year 1905



(see S4, Fig. S6, Fig. S7). Furthermore, the iLand management module ABE was well able to
reproduce the intensification of forest management over the 20[th] century (Fig. S9). Only the
first evaluation period (1924 – 1952) resulted in a small overestimation of simulated harvests.
Further, the simulated wind and bark beetle disturbances between 2007 and 2013 corresponded
well to the expected values derived from KANP inventories (Fig. S10). Our dynamic simulation
approach adequately reproduced the tree species composition and growing stock at the
landscape scale after 95 years of simulation (Fig. S11). Despite an intensification of harvests
until 1997 and the occurrence of a major disturbance event in 1917 – 1923, the average growing
stock on the landscape doubled between 1905 and 2013 (Fig. S12). At the same time total
ecosystem carbon increased by 40.9% (Fig. S13). European beech dominance increased over
the 20[th] century, in particular at lower elevations (Fig. S12, Fig. 1e and 1f). Further details on
historic landscape development can be found in the Supplement S4 and Fig. S5-S13.

## 3.2 Long-term temporal interactions of natural disturbances

We used the empirically derived spatial footprint of two episodes of natural disturbance 90
years apart to investigate the long-term temporal interactions between disturbances. Both
disturbance episodes were found to have a similar impact on growing stock (117,441 m³ and
93,084 m³ of growing stock disturbed at the landscape, respectively), whereas the first episode
affected more than twice the area of the second episode (2334 ha and 1116 ha, respectively).
Only 9.2% of the area disturbed during the first episode was also affected by the second episode
(Fig. 4). Whereas the first disturbance episode mainly affected the central and southern reaches
of the study area, the effects of the second disturbance episode were most pronounced in the
northern parts of the landscape. The odds ratio of 0.49 (p<0.001) revealed a lower probability
that the same location of the first disturbance episode is affected by the second disturbance



episode on the landscape compared to the odds that a previously undisturbed area is disturbed
by the second disturbance episode.

## 3.3 The effect of past disturbance and future climate on 21st century carbon sequestration

Our simulations reveal a considerable impact of past disturbances on the current state of total
ecosystem carbon (Table 1). Simulations without disturbances resulted in an increase in carbon
storage of 43.9 tC ha$^{-1}$ (+11.0%) in 2013 compared to the baseline scenario (i.e., including
natural and human disturbance). The effect of disturbances was strongly dominated by forest
management (97.7%), with only a small influence of the two episodes of natural disturbance.
Past disturbances also resulted in a considerable carbon uptake beyond 2013 (Table 1, Fig. 5,
Fig. 6), *inter alia*, as a result of a persistent recovery of growing stock (Table 2). Past forest
management had a strong and continuous positive legacy effect on the future cumulative carbon
uptake of the landscape (cumulative decrease in NEE until 2099 of -40.6 tC ha$^{-1}$, p<0.001). The
second disturbance episode caused a release of carbon (positive NEE) over the first years of
future simulations, followed by a reversal of the trend towards a negative NEE effect (Fig. S14).
Its overall impact on cumulative NEE at the end of the simulation period was -3.5 tC ha$^{-1}$
(p=0.191), i.e. over the 21st century the recent disturbance period had an overall positive effect
on forest C sequestration. The first disturbance episode had almost no effect on the future
carbon dynamics (NEE effect of -0.2 tC ha$^{-1}$, p=0.792). Simulations of the total legacy effect
of past disturbances (both natural and human) resulted in a cumulative NEE of on average -43.8
tC ha$^{-1}$ (p<0.001) until 2099, indicating that a substantial future C uptake results from the
recovery of forest ecosystems from past disturbance (Fig. 6).





Climate change weakened the carbon sink strength on the landscape, mainly as a result of
climate-mediated differences in successional trajectories of forest ecosystems (Table 2).
However, climate change effects on NEE were more variable compared to disturbance legacy
effects, with increasing uncertainty over time as a result of differences in climate scenarios (Fig.
5). On average, climate change increased the cumulative NEE until 2099 by +24.0 tC ha$^{-1}$
(p<0.001), and thus reduced the carbon uptake of the landscape relative to a continuation of
historic climate (Fig. 6).

## 434  4. Discussion

### 435  4.1 Disturbance interactions in time

Consistent with previous studies assessing the spatial and temporal autocorrelation of
disturbances in Europe (Marini et al., 2012; Schurman et al., 2018; Stadelmann et al., 2013;
Thom et al., 2013), we hypothesized that the disturbance episode in the early 20$^{th}$ century
influenced disturbances in the early 21$^{st}$ century. Our hypothesis was based on the importance
of landscape topography for wind and bark beetle disturbances (Senf and Seidl, 2018; Thom et
al., 2013), and the fact that susceptibility to these agents generally increases with stand age, and
is usually high after 90 years of stand development (Overbeck and Schmidt, 2012; Valinger and
Fridman, 2011). However, our analysis revealed a low probability for the same area to be
affected by the two consecutive disturbance episodes (Fig. 4). This finding is in contrast to
previous studies, which, however, investigated interactions between disturbance events in the
mountain forests of the Alps over only a few years (e.g., Pasztor and others 2014), while we
here analyzed temporal autocorrelation across multiple decades. Furthermore, also our focus on
an entire landscape (and its large heterogeneity in topographic settings and stand conditions) is
different from previous assessments of long-term disturbance feedbacks (but see Hanewinkel



et al., 2008), which have largely focused on plot to stand-level analyses using dendroecology
(e.g., Schurman et al., 2018).
We here tested for an amplifying feedback of natural disturbances in time, expecting high
susceptibility for large parts of the landscape recovering uniformly after the first disturbance
episode, and reaching high susceptibility to wind and bark beetles simultaneously. However,
disturbances can also have negative, dampening effects on future disturbance occurrence, e.g.,
when they lead to increased heterogeneity (Seidl et al., 2016) and trigger autonomous
adaptation of forests to new environmental conditions (Thom et al., 2017c). The low overlap
between the two disturbance episodes reported here could thus be an indication for such a
dampening feedback between disturbances, yet further tests are needed to substantiate this
hypothesis for Central European forest ecosystems. An alternative explanation for the diverging
spatial patterns of the two disturbance episodes might be a different wind direction in the storm
events initiating the two respective episodes, affecting different parts of the highly complex
mountain forest landscapes. Also the legacy effects from past forest management were different
for each episode. The more open structure within stands resulting from heavy exploitation
before 1900 may have increased wind susceptibility in the central and southern reaches of the
landscape regions.

## 4.2 The role of disturbance legacies on future C uptake

Past studies investigating drivers of the forest carbon balance have largely focused either on
historic factors (Keenan et al., 2014; Naudts et al., 2016) or future changes in the environment
(Manusch et al., 2014; Reichstein et al., 2013). Only few studies to date have explicitly
considered disturbance legacies when assessing climate change impacts on the future carbon
uptake of forest ecosystems. However, disregarding legacy effects could lead to a misattribution



of future forest C changes. Here we harnessed an extensive long-term documentation of
disturbance history to study impacts of past disturbance and future climate on the future NEE
of a forest landscape. We found long-lasting legacy effects of past disturbances on the forest
carbon cycle (see also Kashian et al., 2013; Landry et al., 2016; Nunery and Keeton, 2010),
supporting our hypothesis regarding the paramount importance of disturbance legacies for
future C dynamics. In line with a dynamic landscape simulation study for western North
America (Loudermilk et al., 2013) our results revealed that disturbance legacies have a stronger
effect on NEE than changes in climatic conditions (on average 1.7 times higher cumulative
effect over the 21$^{st}$ century – see Fig. 6). Disregarding legacy effects may thus cause a
substantial bias when studying the future carbon dynamics of forest ecosystems. It has to be
noted, however, that over longer future time frames as the one studied here the effects of climate
change will become more important relative to past legacy effects. While we here focused on
the strength of the disturbance legacy effect, future efforts could aim at determining its duration.
Moreover, while our analyses addressed the effects of wind and bark beetles – currently the two
most important natural disturbance agents in Central Europe (Thom et al., 2013) – as well as
their interactions, future climate change may increase the importance of other disturbance
agents not investigated here (see e.g., Wingfield et al., 2017).
The specific disturbance history of our study area, characterized by intensive natural and human
disturbances in the past and major socio-ecological transitions throughout the 20$^{th}$ century, is
key for interpreting our findings. In particular, the cessation of forest management in 1997 had
a very strong impact on the future carbon balance of the landscape (an on average 166.8 times
and 11.5 times higher effect than the first and second episodes of natural disturbances,
respectively – see Fig. 6). In addition to disturbance legacy effects, also climate change
significantly affected the future NEE. In contrast to the general notion that temperate forests
will serve as a strong carbon sink under climate change (Bonan, 2008), our dynamic simulations



suggest that climate change will decrease the ability of the landscape to sequester carbon in the
future, mainly by forcing a transition to forest types with a lower carbon storage potential.
However, considerable uncertainties of climate change impacts on the carbon balance of forest
ecosystems remain (e.g., Manusch et al., 2014). These uncertainties may arise from a wide
range of potential future climate trajectories, but also from a limited understanding of processes
such as the $CO_2$ fertilization effect on forest C uptake (Kroner and Way, 2016; Reyer et al.,
2014). In addition to the direct impacts of climate change (e.g., via temperature and
precipitation changes) on forest ecosystems, climate change will also alter future natural
disturbance regimes (Seidl et al., 2017). The potential for such large pulses of C release from
forests is making the role of forests in climate mitigation strategies highly uncertain (Kurz et
al., 2008; Seidl et al., 2014a).

## 5. Conclusions

Past disturbance (both human and natural) have a long-lasting influence on forest dynamics. In
order to project the future of forest ecosystems we thus need to better understand their past. We
here showed how a combination of historical sources and simulation modeling – applied by an
interdisciplinary team of scientists – can be used to improve our understanding of the long-term
trajectories of forest ecosystems (Bürgi et al., 2017; Collins et al., 2017; Deng and Li, 2016).
Two conclusions can be drawn from the strong historical determination of future forest
dynamics: First, as temperate forests have been managed intensively in many parts of the world
(Deng and Li, 2016; Foster et al., 1998; Naudts et al., 2016), their contribution to climate change
mitigation over the coming decades is likely determined already to a large degree by their past
(see also Schwaab et al., 2015). This means that for the time frame within which a
transformation of human society needs to be achieved in order to retain the earth system within



its planetary boundaries (Steffen et al., 2011), the potential for influencing the role of forests
might be lower than frequently assumed. Efforts to change forest management now to mitigate
climate change through *in situ* C storage, have high potential (Canadell and Raupach, 2008),
but will likely unfold their effects too late to make a major contribution to the transition of the
coming decades. Second, any changes in the disturbance regime of forests – whether intentional
(when altering management) or unintentional in the case of changing natural disturbances –
have profound consequences for the future development of forest ecosystems. This underlines
that a long-term perspective integrating past and future ecosystem dynamics is important when
studying forests, and that decadal to centennial foresight is needed in ecosystem management.

## Author contribution

RS, DT and WR designed the study, RG collected historical data from archives, DT and WR
performed simulations, DT analyzed the outputs, all authors contributed to writing the
manuscript.

## Competing interests

The authors declare that they have no conflict of interest.

## Acknowledgements

This study was supported by the Austrian Climate and Energy Fund ACRP (grant
KR14AC7K11960). W. Rammer and R. Seidl acknowledge further support from the Austrian
Science Fund FWF through START grant Y895-B25. We thank the Austrian Federal Forests



for the permission to access their archives for the collection of the historic data used in this
study. The simulations performed in this study were conducted at the Vienna Scientific Cluster.

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



## Tables


Table 1. Development of total ecosystem carbon stocks (tC ha$^{-1}$) over time and in different scenarios of disturbance history and future climate.
Values are based on iLand simulations and indicate means and standard deviations (SD) over averaged landscape values for the respective
scenarios. "Historic climate" assumes the continuation of the climate 1950 – 2010 throughout the 21$^{st}$ century, while "Climate change" denotes
the effect of three alternative climate change scenarios for the 21$^{st}$ century. The first three columns indicate the respective permutation of the
simulated disturbance history (see also Fig. 3), with the first line representing the historical reconstruction of landscape development. Y=yes,
N=no.

| First nat. dist. episode | Mgmt | Second nat. dist. episode | year 1905 mean | year 1923 mean | SD | year 1997 mean | SD | year 2013 mean | SD | Historic climate year 2099 mean | SD | Climate change year 2099 mean | SD |
|---|---|---|---|---|---|---|---|---|---|---|---|---|---|
| Y | Y | Y | 303.5 | 331.1 | <0.1 | 403.2 | 0.7 | 427.8 | 0.8 | 487.7 | 0.7 | 466.4 | 23.7 |
| Y | N | Y | 303.5 | 331.2 | <0.1 | 457.5 | 0.6 | 466.7 | 0.7 | 487.2 | 1.0 | 463.3 | 20.9 |
| Y | Y | N | 303.5 | 331.0 | <0.1 | 403.2 | 0.7 | 430.6 | 0.7 | 488.2 | 0.7 | 467.0 | 23.3 |
| Y | N | N | 303.5 | 331.2 | <0.1 | 457.5 | 0.5 | 470.9 | 0.7 | 487.3 | 0.7 | 463.4 | 21.1 |
| N | Y | Y | 303.5 | 332.7 | 0.1 | 404.3 | 0.8 | 428.8 | 0.8 | 487.8 | 0.8 | 466.3 | 23.7 |
| N | N | Y | 303.5 | 333.0 | 0.1 | 458.7 | 0.5 | 468.0 | 0.6 | 487.8 | 0.8 | 464.0 | 21.3 |
| N | Y | N | 303.5 | 332.7 | 0.1 | 404.2 | 0.7 | 431.3 | 0.8 | 488.3 | 0.9 | 466.4 | 23.6 |
| N | N | N | 303.5 | 333.0 | 0.1 | 458.6 | 0.5 | 471.7 | 0.6 | 487.9 | 0.9 | 464.1 | 21.0 |





Table 2. Growing stock by tree species (m³ ha⁻¹). Values are based on all iLand simulation runs and indicate species means and standard deviation (SD) over averaged landscape values. "Historic climate" assumes the continuation of the climate 1950 – 2010 throughout the 21st century, while "Climate change" denotes the effect of three alternative climate change scenarios for the 21st century.

| | year 1905 | year 1923 | | year 1997 | | year 2013 | | Historic climate year 2099 | | Climate change year 2099 | |
|---|---|---|---|---|---|---|---|---|---|---|---|
| Tree species | mean | mean | SD | mean | SD | mean | SD | mean | SD | mean | SD |
| *Abies alba* | 4.2 | 2.1 | 0.0 | 9.7 | 2.2 | 12.7 | 2.6 | 28.7 | 6.1 | 33.7 | 7.6 |
| *Fagus sylvatica* | 68.0 | 76.8 | 0.6 | 165.6 | 39.8 | 198.5 | 34.4 | 286.8 | 2.8 | 309.7 | 19.7 |
| *Larix decidua* | 21.5 | 23.9 | 0.2 | 41.7 | 5.2 | 40.5 | 9.7 | 17.4 | 7.9 | 16.2 | 7.1 |
| *Picea abies* | 116.3 | 138.6 | 0.5 | 235.7 | 43.6 | 250.8 | 40.5 | 276.3 | 36.6 | 229.9 | 33.6 |
| Other tree species | 2.3 | 6.0 | 0.2 | 14.7 | 1.4 | 16.0 | 1.6 | 13.4 | 0.5 | 23.8 | 1.7 |
| Total | 212.3 | 247.4 | 0.8 | 467.4 | 79.0 | 518.5 | 66.0 | 622.6 | 35.4 | 613.3 | 46.5 |





## Figures

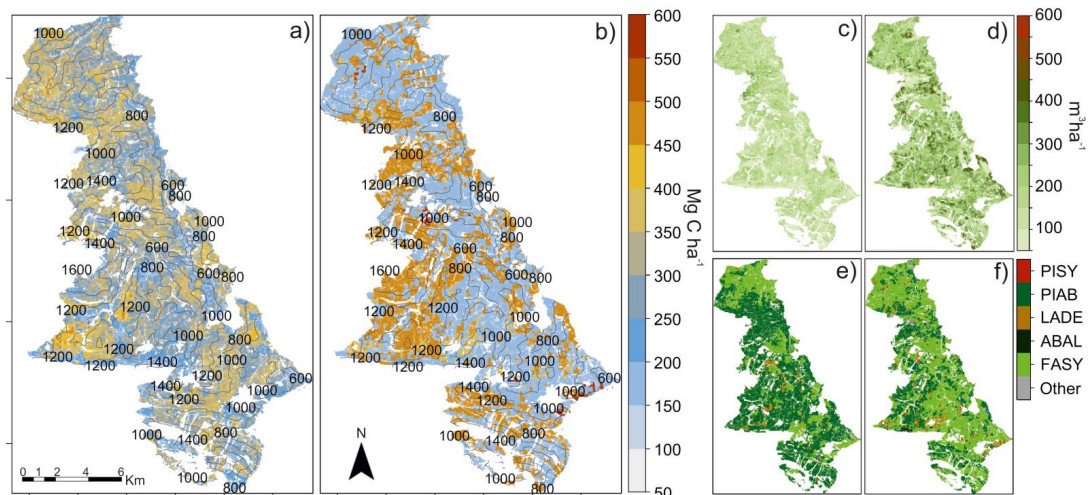

Fig. 1: State of forest ecosystem attributes across the study landscape in 1905 and 2013. (a) and (b) show the distribution of total ecosystem

carbon, (c) and (d) present the growing stock, and (e) and (f) indicate the dominant tree species (i.e., the species with the highest growing stock)

in 1905 and 2013, respectively. PISY = *Pinus sylvestris*, PIAB = *Picea abies*, LADE = *Larix decidua*, ABAL = *Abies alba*, FASY = *Fagus*

*sylvatica*, and "Other" refers to either other dominant species, not individually listed here due to their scarcity, or areas where no trees are

present. Isolines represent elevational gradients.





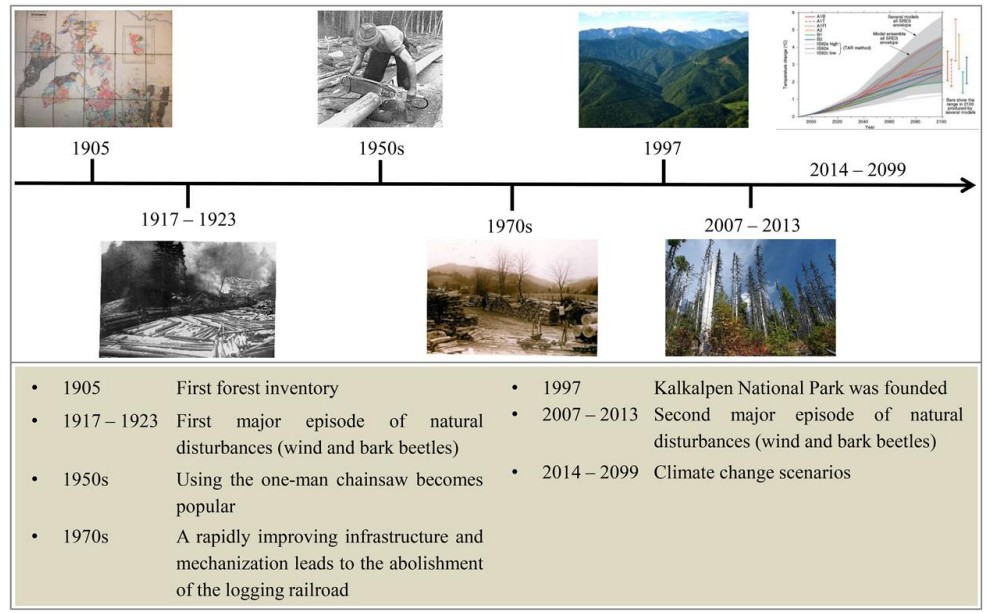


Fig. 2. Timeline of important historic events of relevance for the simulation of the study
landscape. Timeline figures originate from various sources. 1905 and 1917 – 1923: archives of
the Austrian Federal Forests; 1950s: https://waldwissen.at; 1970s: https://atterwiki.at; 1997:
http://kalkalpen.at; 2007 – 2013: photo taken by the authors of this study; 2014 – 2099:
http://climate-scenarios.canada.cau.




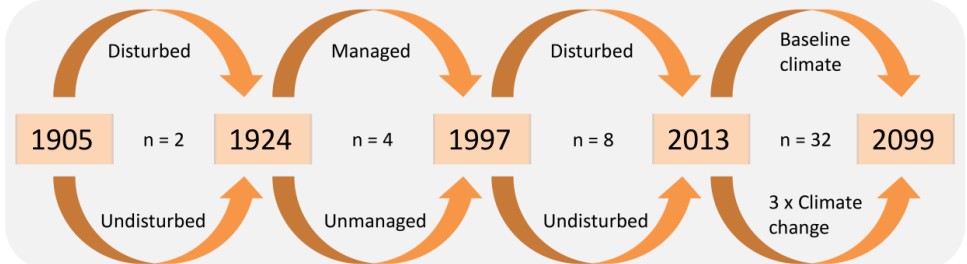


Fig. 3: The disturbance histories and climate futures considered in the simulation. The figure
shows the permutation of factors considered between each time step (years in boxes). n denotes
the number of unique combinations trajectories resulting from the addition of each individual
permutation, each of which was replicated 20 times.



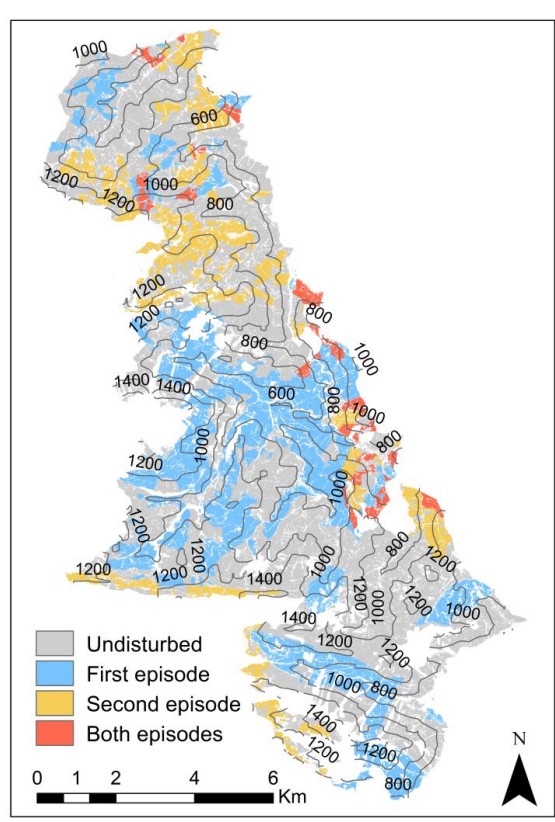


Fig. 4: Disturbance activity in two episodes of natural disturbance 1917 – 1923 (first episode)

and 2007 – 2013 (second episode). Isolines represent elevational gradients.





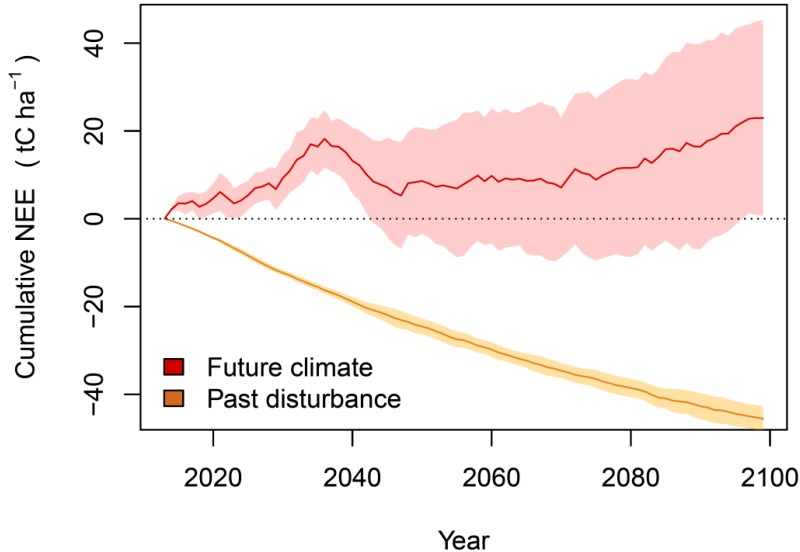


Fig. 5. Mean cumulative change in net ecosystem exchange (NEE) derived by comparing NEE

outputs including past disturbance (historic management and two episodes of natural

disturbance) and future climate with all scenarios excluding past disturbance and baseline

climate, respectively. Shaded areas denote the standard deviation in NEE for the respective

scenarios. NEE is the carbon flux from the ecosystem to the atmosphere.




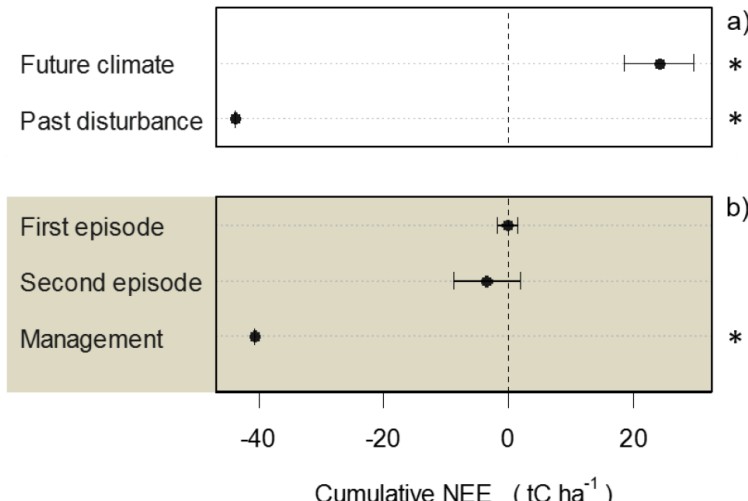

Fig. 6. Effects of future climate and past disturbance on the cumulative NEE of the period 2014 – 2099. a) Effect sizes are calculated from a comparison between climate change and historic climate (both without disturbance) as well as disturbed and undisturbed scenarios (both under historic climate conditions), respectively. Whiskers give the 95% confidence interval around the effect size, and asterisks indicate significant indicators ($\alpha=0.05$). b) In addition to the overall effect of past disturbance, the effect was subdivided into the first and second episodes of natural disturbance as well as human-induced disturbance via management (shaded box).