# Peer review of "Disturbance legacies have a stronger effect on future carbon exchange than climate in a temperate forest landscape Running head: "Disturbance legacies determine C exchange""

_Biogeosciences, 2018_

## Referee Comment (RC1) · Anonymous Referee #1 · 1 May 2018

General comments The manuscript deals with the legacy effects of disturbances (both natural and anthropogenic), and of future climate change, on the C balance of the forest. It is a relevant topic and provides new input to the field. The manuscript is well-written and the work has been done thoroughly.

The first part of the study is an analysis of possible interactions between two past disturbance events. Although I can appreciate the work that has gone into digging out the old archives, my impression is that the analysis was more exploratory in nature, while writing it up, one reference (Schurman et al. 2018) was used as a quick excuse for a hypothesis and the discussion is more focussed to find references on temporal

autocorrelations at different time scales. Perhaps part of the material in the discussion should be transferred to the introduction to provide a more solid hypothesis (like the references in line 442/443), or no hypothesis should be given at all and the patterns found should be discussed against other findings in literature. A weak point here is that there were only two events, and no autocorrelation analysis could be done at different time scales. Furthermore, I'm not always convinced by the arguments the authors bring up in comparing their results to other studies. For example, they state that they find a low probability for the same area to be affected by the two episodes (line 443), which is in contrast to a study that does find correlations between episodes but at very different timescales. I think there is only a contrast if both studies were at the same timescale, and if not, they cannot be compared. Similarly, they state that other studies did find correlations at the plot and stand scale (line 450), but the authors attribute their different finding to the fact that they work at the landscape scale. I do not see why this would yield so different results. If you check a sufficient number of stands and find correlations, I would expect the same would hold true for the landscape. If not, you would expect low correlations at the stand scale as well. Also, lines 457-466 pose some possible reasons why the two events were different. I think they should have enough material to check some of these alternative explanations, or should be able to obtain them with little effort (for example wind direction of both events). Overall, I suggest the authors re-think their hypothesis and discussion for this part of the analysis.

The second part of the study deals with an analysis of the future effect of human and natural disturbances, and future climate change. I think this part of the study is well described and the conclusions are valid. The authors give great care to initialise their model in 1905 using an innovative method, and to simulate the conditions until now, and then project their model into the future. They conclude that the past trajectory is very important to understand the future carbon dynamics. Usually, models would be initialised according to the current state of the forest, and carbon dynamics projected into the future. The current state of the forest would in most cases represent past events, and legacy effects are thus already present. I'm wondering if the 100-year simulation of the past really influences the results, and that this would be a recommended procedure for all models, or that the correct representation of current state and current management is sufficient to include these legacy effects. I could imagine the authors use their new initialisation procedure to represent the current state and compare future projections with and without the 100-year historic run. Perhaps this is too much to add to the current paper, but I would encourage the authors to give some indications on this issue. Are the current initialisation procedures sufficient to take care of past legacies or are longer historic runs needed?

The ordering and numbering of the supplement is a bit strange. S2 and S3 are figures connected to text S1, S4 is text, while S5 and onwards are again figures. While reading the main text, the first reference is S4 while earlier supplementary material is referred to later. Perhaps the supplement could be ordered according to the appearance in the text, and a difference could be made between text and figures.

Specific comments In Figure 1 it would be helpful to add a small map to show where the study area is located within Austria. Line 152: does the model allow for build-up of beetle populations over the years? Line 285: I assume the weather data was adapted to the elevation gradient in the study area somehow? If yes, could you add one sentence about it? Line 356: Simulated species shares were compared against "independent" data for the year 1905. I think 1905 data were used to make the spin-ups for the model. If it is the same data, they are not independent. Or is it really another source? If so, please specify here. Line 410: Is "stock" perhaps better than "storage" here? Line 487: You mention here that you only studied wind and bark beetles, while other agents may become more important in future. I think wildfire was included in your simulations as well. Moreover, you conclude that management was far more important than disturbances, i tihnk this needs to be highlighted here as well.

Technical corrections None.

---

## Referee Comment (RC2) · Anonymous Referee #2 · 2 May 2018

Review for Disturbance legacies have a stronger effect on future carbon exchange than climate in a temperate forest landscape

This study depicts the past and future of a forest landscape in Austria. It aims at evaluating the respective weights of past natural disturbances, past human management, and future climate change on the forest capacity to sequester carbon. For this, the authors reconstructed the landscape history of the federal forest under study using historical data sources. This history is marked by a windstorm in 1905 followed by a bark beetle outbreak, technological evolution of management practices until 1997 when management is ceased, and a second wind and bark beetle event in 2007. The

historical reconstruction results show that there is no correlation between the locations impacted by the first and the second natural disturbance events. In a second time, the authors designed a factorial simulations experiment in which the forest landscape under study undergoes all combinations of conditions : 1917 windstorm and bark beetle event or no, evolution of management practices between 1924 and 1997 or no management after 1924, 1997 windstorm and bark beetle event or no, four climate scenarios from 2013 to 2099. The simulations show that the net ecosystem exchange is dominated by past management found to explain 97.7%. The recovery from past management causes an increase in the future carbon storage. The authors find that by 2100 the effect of human and natural disturbances overcome the effect of climate change.

The object of this study is interesting and timely as the issue of the response of forests to climate change becomes more pressing. The case study is interesting due to its particular history including two large natural disturbance events and a ceasing of human management that allow the analysis of the legacy of management practices on a forest landscape. The simulation experiment is well designed and the model used (iLand) is appropriate to address the questions raised and introduced in a satisfactory way. However, the results and discussion section are somewhat superficial and do miss some important points. Also, the way the study is presented is often confusing or misleading and impairs the comprehension and interpretation of the results. The display items as well as the presentation of the results should be reconsidered to enhance the impact of the work presented.

Detailed comments

Terminology : " disturbance " My main concern is about the use of the word disturbance all along the article, from the title on. The use of this term disturbance is misleading. Usually disturbance refers to natural disturbance (Overpeck et al., 1990; Seidl et al., 2014, 2011). In the present manuscript, it is sometimes used to refer to natural disturbances only (p4 L73 or L395) and sometimes to refer to natural + anthropogenic. It

seems that the authors are aware of the confusion this creates, because most times they explicit that disturbances is natural+anthropogenic (ex : p5L86). Aggregating two very different processes such as management and natural disturbances, on top of being very confusing for the reader, impedes the discussion of one very important result which is the extreme dominance of the effects of management compared to natural disturbances on carbon sequestration of forests. To this regard even the title of the article is misleading or even incorrect since it is not the legacy of the natural disturbance events (explaining only 2,8%) but of past management that has a stronger legacy effect than climate change. The manuscript should be revised to account explicitly for this distinction in the processes analyzed which is obvious in the results.

Methods In the description of the simulation experiment it is noted that each scenario is replicated 20 times (p15 L 347) ? The rationale for this should be explained. What changes between the replicates ? Is there a stochastic component in the model ? L212 : the sentence describing the 1905 age distribution seems a bit far-reaching from fig S8 as the bimodal distribution is not obvious, and the statement is very qualitative.

Results and discussion The manuscript seems very unbalanced with 13,5 pages of intro and methods (both well written and with relevant content) and only 5,5 pages of results and discussion (2,5 and 3 respectively). As reflected by these numbers, the results and discussion sections are sometimes shallow compared to the information presented and the very large number of display items included both in the main text and the supplementary materials (8 and 12 respectively).

Some missing information : - 3.1 Performance of the reconstruction of past events :L377 " a good match" with reference to three supplementary figures, L379 " well able " with reference to one supplementary figure, L381 "small overestimation", L382 "corresponded well" with reference to one supplementary figure etc. all results from section 3.1 are qualitative and based on supplementary figures. An effort should be made to quantify the quality of the reconstruction and to present it in a concise manner in one display item, that, if judged crucial for the validity of the results should be presented in

the main text.

- 3.2 Temporal interaction of disturbance events: the autocorrelation between natural disturbance events is described and found very low. No link is analyzed between disturbance events and management: is there a correlation between stands affected by natural disturbance and species? And age? And density?

- 4.1 The discussion of the lack of autocorrelation between both natural disturbance events and the link to previously published literature is not always clear. For example, the authors state that their hypothesis was that older stands are more prone to wind and bark beetle damages (L442) and link this statement to the low probability of a same area to be affected twice. The fact that a stand is affected by a disturbance does not make it older hence more susceptible to a second disturbance. Several hypotheses are formulated to explain the lack of autocorrelation between both episodes as found in other studies, but none is backed by data so that the discussion is not convincing. One hypothesis is that the longer and larger temporal and spatial scales analyzed here weaken the link found in smaller scale studies. I do not see why stands being more prone would not show up at the landscape scale. Similarly, the hypothesis of a dampening effect of a previous disturbance due to the resulting heterogeneity should be backed by minimal tests on the age and species structures of the affected and non affected stands. As well, the suggestion as to the difference in wind directions of both events needs to be investigated. In summary, an analysis of the characteristics of the stands affected by both natural disturbance events would enlighten this part.

-4.2 disturbance legacies on future C uptake The authors argue that other studies of effect of climate change on carbon sink do not explicitly consider the legacy of past events. It is a bit surprising as past events' legacy in embedded in the initial conditions. The legacy spinup method derived here is interesting and relevant but should be placed in the context of alternative methods to describe forest initial conditions, see for example (Crookston et al., 2010; Garcia-Gonzalo et al., 2007; Hurtt et al., 2002; Karjalainen et al., 2002; Peng et al., 2009). The novelty of this study does not seem to be the

inclusion of the disturbances' legacy but their quantification so this section should be rephrased. Several sentences are not backed by any reference and should be justified and developed. For example on L484, the sentence stating that these results may not hold for longer time frames, on L499 the sentence interepreting the simulation results as a change in forest types.

- effect of climate change It is not explained in many details what response of forest growth to climate change is simulated by iLand (with respect to species or altitude for example). The results shown here on the comparison of climate change and management are highly related to the processes included in the modeling exercise as correctly stated in L501-507 and would deserve a more in-depth explanation. A discussion section on the simulated response of forest growth to climate change only would help put the results in perspective.

Display items Some display items do not help the understanding of the text, are redundant, or at the contrary lack information, and so should be rethought as material that supports the claim made in the text.

Fig2 aims at summarizing the events included in the historical reconstruction of the forest landscape. Its design is more appropriate for a slideshow than a written article. Fig3 illustrates how the events shown in fig2 are included in the simulation experiments. Its design is confusing, especially with the 'n' that is cumulated from left to right (it takes some time to understand this) and that attempts at expliciting the factorial combination of the events simulated. These 2 figures could be condensed into a single display item where only the information relevant to the study would be presented. For example a table structured as below:

Period / Scenarios' options / details

1905-1924 / disturbed / storm+bark beetle+. . .

/ undisturbed /
1924-1997 / managed / Technological improvements

/ unmanaged / Forest left to grow

1997-2013 / disturbed /

/ undisturbed /

2013-2099 / Climate scenario 1 /

/ Climate scenario 2 /

/ Climate scenario 3 /

/ Climate scenario 4 /

Other problematic display items are Fig5, Fig6 and FigS14. These three figures are redundant and should be combined into a single figure that shows the time evolution of NEE attributed to climate, event1, event2, and management. Please explain 'cumulative NEE'. From fig5, since the climate driven cumulative NEE decreases it means that the forest becomes a source of carbon between 2035 and 2050? This pattern should be discussed (see comment on 'effect of climate change' ).

Supplementary materials The supplementary figures are excessive. Some could be merged into a single figure such as Fig. S11 and S12 that show the same variable (growing stock per species) Some are not even cited in the text such as Fig S13. Fig. S5 is not clear, why showing two sites in the fictitious landscape map with only on stand development below. Letters A to D are shown but not used in the explanation but the outcome of the spinup (letter D I guess?) is not highlighted.

Technical details P3 L49 'Keenan and others' instead of 'et al' the numeration of the supplementary materials is confusing with only one line of numbering for text sections and figures. There should be section S1, section S2, section S3, figure S1, figure S2, figure S3, figure S4. . . Crookston, N.L., Rehfeldt, G.E., Dixon, G.E., Weiskittel, A.R., 2010. Addressing climate change in the forest vegetation simulator to assess impacts

on landscape forest dynamics. For. Ecol. Manag. 260, 1198–1211. Garcia-Gonzalo, J., Peltola, H., Gerendiain, A.Z., Kellomäki, S., 2007. Impacts of forest landscape structure and management on timber production and carbon stocks in the boreal forest ecosystem under changing climate. For. Ecol. Manag. 241, 243–257. Hurtt, G.C., Pacala, S.W., Moorcroft, P.R., Caspersen, J., Shevliakova, E., Houghton, R.A., Moore, B., 2002. Projecting the future of the US carbon sink. Proc. Natl. Acad. Sci. 99, 1389–1394. Karjalainen, T., Pussinen, A., Liski, J., Nabuurs, G.-J., Erhard, M., Eggers, T., Sonntag, M., Mohren, G.M.J., 2002. An approach towards an estimate of the impact of forest management and climate change on the European forest sector carbon budget: Germany as a case study. For. Ecol. Manag. 162, 87–103. Overpeck, J.T., Rind, D., Goldberg, R., 1990. Climate-induced changes in forest disturbance and vegetation. Nature 343, 51. Peng, C., Zhou, X., Zhao, S., Wang, X., Zhu, B., Piao, S., Fang, J., 2009. Quantifying the response of forest carbon balance to future climate change in Northeastern China: model validation and prediction. Glob. Planet. Change 66, 179–194. Seidl, R., Schelhaas, M.-J., Lexer, M.J., 2011. Unraveling the drivers of intensifying forest disturbance regimes in Europe. Glob. Change Biol. 17, 2842–2852. Seidl, R., Schelhaas, M.-J., Rammer, W., Verkerk, P.J., 2014. Increasing forest disturbances in Europe and their impact on carbon storage. Nat. Clim. Change 4, 806.

Please also note the supplement to this comment:
https://www.biogeosciences-discuss.net/bg-2018-145/bg-2018-145-RC2-supplement.pdf
* * *

---

## Referee Comment (RC3) · Anonymous Referee #3 · 4 May 2018

General comments : The research article named 'Disturbance legacies have a stronger effect on future carbon exchange than climate in a temperate forest landscape,' try to explore the effect of disturbance legacies and climate change in the projection of the forest carbon sequestration. In order to do that, they reconstruct a well documented historical scenario of an Austrian forest landscape with two disturbance events and one forest management shift. At the end of the paper, they encourage the scientific community to take into account the forest history when initializing the forest state before running projections of the forest dynamic. This is a nice attempt to promote the integration of disturbances and abrupt mortality in model development. I really appreciate the quality of the work done by the simulation experiment and the past reconstruction

forest state with the new method of spin-up. I am convinced that this paper can be published without deep changes in the structure and the content. However, five points need to be clarified:

1) The results of the simulation experiment show that past forest management (absence or presence) is the main factor to explain the divergence between simulations. But this finding is not central to the paper! Instead of that, the authors define forest management as a human disturbance (that is perfectly true) and merged natural and human disturbances in one general disturbance term. This merging leads to a mis-interpretation of the title and the conclusion because, for most of the ecologist and the forest manager, disturbance legacies always refer to an extreme event legacy like storms, beetles outbreaks, fires or droughts. My advice is to explicitly divide interpretation of the result into the natural and the human disturbance. For example, the title will become: "Human disturbance/forest management/human activity legacies have a stronger effect on future carbon exchange than climate in a temperate forest landscape."

2) the authors need to be careful with the last statement of the title: "than climate in a temperate forest landscape" because the authors only realized simulations with a medium climate change scenario (A1B). The strongest climate change scenario like the RCP 8.5 is most likely to happen, and it will have a stronger impact. In addition, the authors forget to take into account the indirect effect of CC on forest growth via the increase of the frequencies and the intensities of the extreme events. This partly due to the setup of the simulation experiment where disturbances are forced and disconnected to the mortality module of iland. But this interaction can be simulated in iland because the authors already developed abrupt mortality module into this model.

3) The way the authors display the results of the simulation experiment is very confusing. The figure 5 for example which display the difference between reference NEE and alternative NEE, starts to diverge from 2013 and not from 1905. The simulations without management should not be far from other simulation in 2013?

4) In table 1, we can see a difference of about 40 tC ha between managed and unmanaged simulations. The strangest thing here is that in 2099 this difference disappears (compensation process?). This is interesting but the authors don't mention that in the discussion. Why? and why the figure 5 doesn't display that?

5) Did the two imposed disturbances have a different impact on the forest across simulations? If not, it means that the authors can't observe the legacy effect of one disturbance to another future one. It is maybe the reason why they don't observe a strong effect of natural disturbances. Due to this lake of interaction, the interesting questions like: - Can this forest have the capacity to absorb extreme events well enough to keep the same level of NEE if the intensity and the frequencies of natural disturbances will increase? Or - Are the forest management made between 1905 and 1997 is able to change disturbance impact on NEE in the future? cannot be tackled. It is a pity because it will strengthen the purpose of this paper.

---

## Author Comment (AC1) · 29 May 2018

REFEREE #1: General comments The manuscript deals with the legacy effects of disturbances (both natural and anthropogenic), and of future climate change, on the C balance of the forest. It is a relevant topic and provides new input to the field. The manuscript is well-written and the work has been done thoroughly.

AUTHORS: We thank the referee for his/her overall positive evaluation of our study.

REFEREE #1: The first part of the study is an analysis of possible interactions between

two past disturbance events. Although I can appreciate the work that has gone into digging out the old archives, my impression is that the analysis was more exploratory in nature, while writing it up, one reference (Schurman et al. 2018) was used as a quick excuse for a hypothesis and the discussion is more focussed to find references on temporal autocorrelations at different time scales. Perhaps part of the material in the discussion should be transferred to the introduction to provide a more solid hypothesis (like the references in line 442/443), or no hypothesis should be given at all and the patterns found should be discussed against other findings in literature. A weak point here is that there were only two events, and no autocorrelation analysis could be done at different time scales. Furthermore, I'm not always convinced by the arguments the authors bring up in comparing their results to other studies. For example, they state that they find a low probability for the same area to be affected by the two episodes (line 443), which is in contrast to a study that does find correlations between episodes but at very different timescales. I think there is only a contrast if both studies were at the same timescale, and if not, they cannot be compared. Similarly, they state that other studies did find correlations at the plot and stand scale (line 450), but the authors attribute their different finding to the fact that they work at the landscape scale. I do not see why this would yield so different results. If you check a sufficient number of stands and find correlations, I would expect the same would hold true for the landscape. If not, you would expect low correlations at the stand scale as well. Also, lines 457-466 pose some possible reasons why the two events were different. I think they should have enough material to check some of these alternative explanations, or should be able to obtain them with little effort (for example wind direction of both events). Overall, I suggest the authors re-think their hypothesis and discussion for this part of the analysis.

AUTHORS: The referee makes an important point with regard to revising the hypotheses and the part of the discussion pertaining to the first part of our analysis. Given the lack of explicit data on past disturbance episodes comparisons as the one undertaken here are rare, which makes embedding it in the literature challenging. Furthermore, as some important characteristics of the first disturbance episode remain unknown (e.g.,

exact wind speed, wind direction) some uncertainties about the causes of the difference between the two episodes will necessarily have to remain. Furthermore, we'd like to point out that an analysis of individual drivers of the Central European disturbance regime is beyond the scope of the current contribution, in fact the causes of natural disturbances have been investigated in detail already in prior studies (Marini et al., 2012; Overbeck and Schmidt, 2012; Pasztor et al., 2014, 2015; Thom et al., 2013). Nonetheless, we agree with Reviewer #1 that the correlation between the two disturbance events warrants further attention. Congruent with the suggestions of referee #1 and referee #2, we will add another analysis to investigate the contribution of forest management to the second disturbance episode. Based on our factorial simulation design, we will analyze the effect of four different combinations of previous natural disturbances and management on the second disturbance episode in 320 simulations (those including the second disturbance episode). This analysis will provide further insights into the drivers of the second disturbance episode. Following the referee's advice we will also reformulate our hypothesis and substantiate it with some of the material provided in the discussion section. Based on the results of the new analysis, we will also adapt the discussion in section 4.1.

REFEREE #1: The second part of the study deals with an analysis of the future effect of human and natural disturbances, and future climate change. I think this part of the study is well described and the conclusions are valid. The authors give great care to initialise their model in 1905 using an innovative method, and to simulate the conditions until now, and then project their model into the future. They conclude that the past trajectory is very important to understand the future carbon dynamics. Usually, models would be initialised according to the current state of the forest, and carbon dynamics projected into the future. The current state of the forest would in most cases represent past events, and legacy effects are thus already present. I'm wondering if the 100-year simulation of the past really influences the results, and that this would be a recommended procedure for all models, or that the correct representation of current state and current management is sufficient to include these legacy effects. I could imagine

the authors use their new initialisation procedure to represent the current state and compare future projections with and without the 100-year historic run. Perhaps this is too much to add to the current paper, but I would encourage the authors to give some indications on this issue. Are the current initialisation procedures sufficient to take care of past legacies or are longer historic runs needed?

AUTHORS: We thank the referee for the positive evaluation of the second part of our study. The Reviewer is in fact correct in stating that usually legacy information is captured via the initialization of a model. This is not different in iLand, the model applied here. However, our point here is a slightly different one, namely: How different would the state of the forest (and hence the initialization of a simulation model) be if it would have had a different disturbance history? We thus quantify the structural effects of different past activities onto the state of the forest in 2013, and investigate how long these differences persist into the future, given everything else is equal. So the Reviewer is correct in assuming that if the initial conditions are known the legacies are adequately captured for modeling. However, in many cases the initial conditions of a forest landscape are incompletely known. This is for instance the case for the state of our landscape in 1905, for which we have information about species composition and growing stock, but not for other important variables (e.g., soil C pools, the spatial composition and configuration of stands). The legacy spin-up approach presented here was designed to address this very issue. In the revision we will revisit our description of legacy effects and reword/ amend it where necessary, in order to make explicitly clear what our contribution here is, and avoid confusions for the reader.

REFEREE #1: The ordering and numbering of the supplement is a bit strange. S2 and S3 are figures connected to text S1, S4 is text, while S5 and onwards are again figures. While reading the main text, the first reference is S4 while earlier supplementary material is referred to later. Perhaps the supplement could be ordered according to the appearance in the text, and a difference could be made between text and figures.

AUTHORS: We agree that the enumeration of the supplement can be improved. This

has also been suggested by referee #2. In our revision, we will restructure the supplement into sections (e.g., Section S1) and provide all figure with a consecutive number. We will take care that sections and figures will be numbered continuously throughout the text.

REFEREE #1: Specific comments

In Figure 1 it would be helpful to add a small map to show where the study area is located within Austria.

AUTHORS: We will complement the figure with an insert showing a map of Austria and the location of the landscape.

REFEREE #1: Line 152: does the model allow for build-up of beetle populations over the years?

AUTHORS: Yes indeed, the process-based bark beetle module implemented in iLand is able to simulate the build-up of bark beetle populations over years. Weather conditions affect the bark beetle population directly (e.g., the number of generations and sister broods per year, as well as winter survival rate). Furthermore, the vitality of trees and thus their defense capacity (simulated via carbon starvation) as well as the amount of windthrown tree (easily colonizable breeding material) influence beetle populations in the model. Seidl and Rammer (2017) found that iLand was well able to reproduce the 2nd bark beetle disturbance episode contained in our analysis here. We will add this information in the revision of the manuscript.

REFEREE #1: Line 285: I assume the weather data was adapted to the elevation gradient in the study area somehow? If yes, could you add one sentence about it?

AUTHORS: Indeed, elevation gradients are captured in the climate data used. The climate data from 1950 – 2099 were all statistically downscaled to a resolution of 100 x 100 m by means of quantile mapping. For the years 1905 – 1949, we had only temperature and precipitation from a nearby weather station. We thus drew a climate

from the period 1950 – 2099 for each missing year by matching its temperature and precipitation data to that of the weather station record for 1905 – 1949. We will clarify in this approach in the revised version of the manuscript.

REFEREE #1: Line 356: Simulated species shares were compared against "independent" data for the year 1905. I think 1905 data were used to make the spin-ups for the model. If it is the same data, they are not independent. Or is it really another source? If so, please specify here.

AUTHORS: We agree with the referee that the comparison of the simulation with the observed data is not entirely independent, as the observed data was used to guide the spin-up procedure. We will change the text accordingly in the revised version of the manuscript.

REFEREE #1: Line 410: Is "stock" perhaps better than "storage" here?

AUTHORS: We will change to "stock" in the revision.

REFEREE #1: Line 487: You mention here that you only studied wind and bark beetles, while other agents may become more important in future. I think wildfire was included in your simulations as well. Moreover, you conclude that management was far more important than disturbances, i tihnk this needs to be highlighted here as well.

AUTHORS: Our simulations included disturbances from wind and bark beetles only as stated in l. 143 – 146 "As wind and bark beetles are of paramount importance for the past and future disturbance regimes of Central Europe's forests (Seidl et al., 2014a; Thom et al., 2013), we employed only these two process-based disturbance submodules in our simulations". Although it is correct that iLand is able to simulate disturbance from wildfire, we did not include wildfires here as they are not an important component of the disturbance regime in our study system. With regard to management we agree on its importance, and will highlight this more explicitly in the revision (see also our response to similar suggestions by the other Reviewers).

References Marini, L., Ayres, M. P., Battisti, A. and Faccoli, M.: Climate affects severity and altitudinal distribution of outbreaks in an eruptive bark beetle, Clim. Change, 115(2), 327–341, doi:10.1007/s10584-012-0463-z, 2012. Overbeck, M. and Schmidt, M.: Modelling infestation risk of Norway spruce by Ips typographus (L.) in the Lower Saxon Harz Mountains (Germany), For. Ecol. Manage., 266, 115–125, doi:10.1016/j.foreco.2011.11.011, 2012. Pasztor, F., Matulla, C., Rammer, W. and Lexer, M. J.: Drivers of the bark beetle disturbance regime in Alpine forests in Austria, For. Ecol. Manage., 318, 349–358, doi:10.1016/j.foreco.2014.01.044, 2014. Pasztor, F., Matulla, C., Zuvela-Aloise, M., Rammer, W. and Lexer, M. J.: Developing predictive models of wind damage in Austrian forests, Ann. For. Sci., 72(3), 289–301, doi:10.1007/s13595-014-0386-0, 2015. Seidl, R. and Rammer, W.: Climate change amplifies the interactions between wind and bark beetle disturbances in forest landscapes, Landsc. Ecol., 32(7), 1485–1498, doi:10.1007/s10980-016-0396-4, 2017. Thom, D., Seidl, R., Steyrer, G., Krehan, H. and Formayer, H.: Slow and fast drivers of the natural disturbance regime in Central European forest ecosystems, For. Ecol. Manage., 307, 293–302, doi:10.1016/j.foreco.2013.07.017, 2013.

---

## Author Comment (AC2) · 29 May 2018

REFEREE #2: This study depicts the past and future of a forest landscape in Austria. It aims at evaluating the respective weights of past natural disturbances, past human management, and future climate change on the forest capacity to sequester carbon. For this, the authors reconstructed the landscape history of the federal forest under study using historical data sources. This history is marked by a windstorm in 1905 followed by a bark beetle outbreak, technological evolution of management practices until 1997 when management is ceased, and a second wind and bark beetle event in

2007. The historical reconstruction results show that there is no correlation between the locations impacted by the first and the second natural disturbance events. In a second time, the authors designed a factorial simulations experiment in which the forest landscape under study undergoes all combinations of conditions : 1917 windstorm and bark beetle event or no, evolution of management practices between 1924 and 1997 or no management after 1924, 1997 windstorm and bark beetle event or no, four climate scenarios from 2013 to 2099. The simulations show that the net ecosystem exchange is dominated by past management found to explain 97.7%. The recovery from past management causes an increase in the future carbon storage. The authors find that by 2100 the effect of human and natural disturbances overcome the effect of climate change. The object of this study is interesting and timely as the issue of the response of forests to climate change becomes more pressing. The case study is interesting due to its particular history including two large natural disturbance events and a ceasing of human management that allow the analysis of the legacy of management practices on a forest landscape. The simulation experiment is well designed and the model used (iLand) is appropriate to address the questions raised and introduced in a satisfactory way. However, the results and discussion section are somewhat superficial and do miss some important points. Also, the way the study is presented is often confusing or misleading and impairs the comprehension and interpretation of the results. The display items as well as the presentation of the results should be reconsidered to enhance the impact of the work presented.

AUTHORS: We thank the referee for his/her interest in our study and the very thoughtful review with valuable comments to help us improve our manuscript. In the revision, we will focus on dissolving the confusing interpretation of results that were highlighted by the reviewer. See our responses below on how we plan to achieve this.

REFEREE #2: Detailed comments Terminology : " disturbance " My main concern is about the use of the word disturbance all along the article, from the title on. The use of this term disturbance is misleading. Usually disturbance refers to natural disturbance

(Overpeck et al., 1990; Seidl et al. 2014, 2011). In the present manuscript, it is sometimes used to refer to natural disturbances only (p4 L73 or L395) and sometimes to refer to natural + anthropogenic. It seems that the authors are aware of the confusion this creates, because most times they explicit that disturbances is natural+anthropogenic (ex : p5L86). Aggregating two very different processes such as management and natural disturbances, on top of being very confusing for the reader, impedes the discussion of one very important result which is the extreme dominance of the effects of management compared to natural disturbances on carbon sequestration of forests. To this regard even the title of the article is misleading or even incorrect since it is not the legacy of the natural disturbance events (explaining only 2,8%) but of past management that has a stronger legacy effect than climate change. The manuscript should be revised to account explicitly for this distinction in the processes analyzed which is obvious in the results.

AUTHORS: We thank the Reviewer for pointing this out. Our idea in the initial submission was to first combine natural and human disturbances to quantify the overall disturbance effect on carbon storage, and subsequently disentangle the partial effects of natural and human disturbances. As two of the three referees (referee #2 and referee #3) found the combination of natural and human disturbances into the overall disturbance effect confusing and problematic, we concede that this idea needs to be revised. In the revision we will clearly distinguish between management and natural disturbances throughout our study. We will rephrase the title of the study into "Legacies of forest management have a stronger effect on future carbon exchange than climate and natural disturbances in a temperate forest landscape".

REFEREE #2: Methods In the description of the simulation experiment it is noted that each scenario is replicated 20 times (p15 L 347) ? The rationale for this should be explained. What changes between the replicates ? Is there a stochastic component in the model ?

AUTHORS: iLand is a process-based model including fully-dynamic submodules for

natural disturbances and forest management, and each of these components contain stochasticity (e.g., the spread of an individual bark beetle cohort from an infested tree is determined by drawing from a distribution of empirically determined dispersal distances, with spread distance drawn randomly between $0°$ and $360°$). To account for this stochasticity, we have replicated every simulation 20 times. This particular number has been proven to be a good middle ground between determining robust results and keeping simulation times reasonable in previous applications of the model (e.g., Seidl et al., 2018; Thom et al., 2017). We will explain the rationale of the replicates more explicitly in the revised version of the manuscript.

REFEREE #2: L212: the sentence describing the 1905 age distribution seems a bit far-reaching from fig S8 as the bimodal distribution is not obvious, and the statement is very qualitative.

AUTHORS: We agree with the referee and will change the text accordingly.

REFEREE #2: Results and discussion The manuscript seems very unbalanced with 13,5 pages of intro and methods (both well written and with relevant content) and only 5,5 pages of results and discussion (2,5 and 3 respectively). As reflected by these numbers, the results and discussion sections are sometimes shallow compared to the information presented and the very large number of display items included both in the main text and the supplementary materials (8 and 12 respectively).

AUTHORS: We only partly agree with the referee on this point. We feel that it is important to include an extensive methods section in highly complex and computational extensive studies in order to ensure the highest possible degree of reproducibility (Scheller et al., 2011; Schwaab et al., 2015; Temperli et al., 2013). As 5 of the 8 figures as well as both tables are anchored in the results section, the manuscript is overall less imbalanced as it may seem based on text pages only. Moreover, besides the 5.5 pages for results and discussion, there is another page of text making up the conclusion section, which should be considered as well. With regard to the supplement we feel that

the extensive additional material presented here helps the reader to understand our study and provides additional context on the validation and applicability of the methods used in our study (while not further clogging the main text). Nonetheless, we will further strengthen the results and discussion sections in the revised version of the manuscript through an additional analysis of the impacts of management and the first disturbance episode on the second disturbance episode (see our response below). Additionally, we will improve the clarity of information provided in the results and discussion sections based on the reviewer comments. However, we will refrain from omitting parts of the methods (which the referee agrees are relevant to understand the study) or prolong the results and discussion sections extensively (as our manuscript is already fairly long).

REFEREE #2: Some missing information : - 3.1 Performance of the reconstruction of past events:L377 " a good match" with reference to three supplementary figures, L379 " well able " with reference to one supplementary figure, L381 "small overestimation", L382 "corresponded well" with reference to one supplementary figure etc. all results from section 3.1 are qualitative and based on supplementary figures. An effort should be made to quantify the quality of the reconstruction and to present it in a concise manner in one display item, that, if judged crucial for the validity of the results should be presented in the main text.

AUTHORS: We understand the desire of Referee #2 for a single, concise evaluation result. However, we here follow a patter-oriented modeling approach (Grimm, 2005), which means that a variety of very different indicators are in order to evaluate the model's ability to reproduce the empirically derived historic data (i.e., tree species composition in 1905 and 1999, management, natural disturbances, carbon storage). In our opinion these cannot be combined into a single number/ figure, as such a combination may hide important information regarding model performance (e.g., the model could be doing very well wrt one indicator while performing poorly wrt a second one, which would give on average moderate performance; if the poorly captured indicator is, however, of particular importance for the study, this information would be lost in such an

aggregate evaluation). After careful consideration of the Referees comment we thus have decided to retain the multidimensional nature of our evaluation. In the revision we will explain this in more detail and provide our rationale for this approach for the reader.

REFEREE #2: - 3.2 Temporal interaction of disturbance events: the autocorrelation between natural disturbance events is described and found very low. No link is analyzed between disturbance events and management: is there a correlation between stands affected by natural disturbance and species? And age? And density?

AUTHORS: We thank the Referee for bringing up the issue of management in this context. In fact the possibility of a connection between management and the second disturbance episode has also been pointed out by referee #1, and we agree that this is an important issue here. We thus suggest to add an additional analysis in this regard. Following the advice of referees #1 and #2, we will investigate the contribution of forest management on the second disturbance episode in our revision. We will analyze the effect of all 4 potential combinations of previous natural disturbances and management on the second disturbance episode in 320 simulations (i.e., those including the second disturbance episode). This additional analysis will help us to investigate legacy effects on disturbances, and give further insights into the first disturbance episode on the second episode, as well as the effect of forest management on the second disturbance episode. We feel that such an additional analysis will considerably strengthen our submission further, and thank the Referee for suggesting it!

REFEREE #2: - 4.1 The discussion of the lack of autocorrelation between both natural disturbance events and the link to previously published literature is not always clear. For example, the authors state that their hypothesis was that older stands are more prone to wind and bark beetle damages (L442) and link this statement to the low probability of a same area to be affected twice. The fact that a stand is affected by a disturbance does not make it older hence more susceptible to a second disturbance. Several hypotheses are formulated to explain the lack of autocorrelation between both episodes as found in other studies, but none is backed by data so that the discussion is not convincing.

One hypothesis is that the longer and larger temporal and spatial scales analyzed here weaken the link found in smaller scale studies. I do not see why stands being more prone would not show up at the landscape scale. Similarly, the hypothesis of a dampening effect of a previous disturbance due to the resulting heterogeneity should be backed by minimal tests on the age and species structures of the affected and non affected stands. As well, the suggestion as to the difference in wind directions of both events needs to be investigated. In summary, an analysis of the characteristics of the stands affected by both natural disturbance events would enlighten this part.

AUTHORS: As highlighted by referees #1 and #2 we agree that this part of the discussion needs to be revised. We have tried to find other studies investigating the spatial autocorrelation of two consecutive major disturbance episode, but spatio-temporal autocorrelation of disturbances has been usually either described over very limited time frames (e.g., Pasztor et al., 2014) or the spatial resolution for the comparison of disturbances over longer time frames has been very coarse (e.g., Senf and Seidl, 2018). In this regard, our analysis constitutes a novel contribution, improving our understanding of disturbance dynamics over extended temporal scales. Although we have spatially explicit disturbance data for both events, we cannot conduct a process-based analysis at the level of individual drivers. The reason is that we do not know all the characteristics of the wind event of the 1917-1923 disturbance episode (e.g., wind direction and wind speeds) as these have not been faithfully documented. Moreover, we feel that the analysis of disturbance drivers is beyond the scope of the current contribution, as this has been investigated in more detail in other studies in Central European ecosystems (e.g., Marini et al., 2012; Overbeck and Schmidt, 2012; Pasztor et al., 2014, 2015; Thom et al., 2013). Nonetheless, we will improve the analysis of how past legacies have affected recent disturbances in the revised version of the manuscript. As mentioned above we will add a new analysis investigating the contribution of the first disturbance episode and forest management on the second disturbance episode. This analysis will serve to substantiate our finding of a weak contribution of one disturbance episode on the other, and provides more insights into the effect of forest management

on the Central European disturbance regime. Based on these results we will improve the discussion in section 4.1 of the revised manuscript.

REFEREE #2: -4.2 disturbance legacies on future C uptake The authors argue that other studies of effect of climate change on carbon sink do not explicitly consider the legacy of past events. It is a bit surprising as past events' legacy in embedded in the initial conditions. The legacy spinup method derived here is interesting and relevant but should be placed in the context of alternative methods to describe forest initial conditions, see for example (Crookston et al., 2010; Garcia-Gonzalo et al., 2007; Hurtt et al., 2002; Karjalainen et al., 2002; Peng et al., 2009). The novelty of this study does not seem to be the inclusion of the disturbances' legacy but their quantification so this section should be rephrased. Several sentences are not backed by any reference and should be justified and developed. For example on L484, the sentence stating that these results may not hold for longer time frames, on L499 the sentence interepreting the simulation results as a change in forest types.

AUTHORS: We agree with the reviewer that the legacy effects are indeed embedded in the initial conditions, if the initialization is based on a comprehensive set of empirical data. It is also correct that the quantification of the legacy effect is the actual novel contribution of our study (see also our response to a similar comment of Referee #1). We will rephrase this section accordingly, and add references as suggested. We will also discuss the legacy spin-up method in the context of other established methods.

REFEREE #2: - effect of climate change It is not explained in many details what response of forest growth to climate change is simulated by iLand (with respect to species or altitude for example). The results shown here on the comparison of climate change and management are highly related to the processes included in the modeling exercise as correctly stated in L501-507 and would deserve a more in-depth explanation. A discussion section on the simulated response of forest growth to climate change only would help put the results in perspective.

AUTHORS: We agree that this is important information for readers in order to understand the results presented here. iLand considers both direct and indirect vegetation responses to climate change. For instance, temperature increases directly affect processes such as leaf phenology and the length of vegetation period, the efficiency of photosynthesis (modeled using a state acclimation approach following Mäkelä et al. 2008), and the availability of water in the soil (via altered evapotranspiration rates). Similarly, rising CO2 levels directly affect net primary production via CO2 fertilization. Thus, climate change might affect growth of one species differently than that of another species (direct effect), leading to a change in forest competition and structure (indirect effect). We will provide more details of the climate change effects on forest vegetation in iLand in the revised methods section. We will also explicitly refer the reader to the more technical iLand papers describing this issue (Seidl et al., 2012b, 2012a; Thom et al., 2017).

REFEREE #2: Display items Some display items do not help the understanding of the text, are redundant, or at the contrary lack information, and so should be rethought as material that supports the claim made in the text. Fig2 aims at summarizing the events included in the historical reconstruction of the forest landscape. Its design is more appropriate for a slideshow than a written article. Fig3 illustrates how the events shown in fig2 are included in the simulation experiments. Its design is confusing, especially with the 'n' that is cumulated from left to right (it takes some time to understand this) and that attempts at expliciting the factorial combination of the events simulated. These 2 figures could be condensed into a single display item where only the information relevant to the study would be presented. For example a table structured as below: Period / Scenarios' options / details 1905-1924 / disturbed / storm+bark beetle+ ... / undisturbed / 1924-1997 / managed / Technological improvements / unmanaged / Forest left to grow 1997-2013 / disturbed / / undisturbed / 2013-2099 / Climate scenario 1 / / Climate scenario 2 / / Climate scenario 3 / / Climate scenario 4 /

AUTHORS: As mentioned by the Referee, Figures 2 and 3 are highlighting two different

aspects of the study: Figure 2 represents the history of events on the landscape while Figure 3 shows the simulation design. We will explore combining these aspects into a single figure as suggested. Another option is combining the information into a single table, as suggested. We will revise our display items for clarity and remove redundancy where possible in the revised version of the manuscript.

REFEREE #2: Other problematic display items are Fig5, Fig6 and FigS14. These three figures are redundant and should be combined into a single figure that shows the time evolution of NEE attributed to climate, event1, event2, and management. Please explain 'cumulative NEE'. From fig5, since the climate driven cumulative NEE decreases it means that the forest becomes a source of carbon between 2035 and 2050? This pattern should be discussed (see comment on 'effect of climate change' ).

AUTHORS: The referee is right that there is some redundancy between these figures as the endpoints in Fig. 5 and Fig. S14 reflect the effect size in Fig. 6. Also with regard to the previous comment to distinguish between management and natural disturbance throughout the paper, it makes sense to combine these figures, and we will make these changes in the revised version of the manuscript. However, the interpretation of NEE by the Referee is not correct here. As NEE = -NEP an decrease in NEE means an increase of carbon in terrestrial ecosystems, i.e., between 2035 and 2050 there is an uptake of carbon by forests under climate change. We have provided a definition of NEE in l. 363f.: "NEE denotes the net C flux from the ecosystem to the atmosphere, with negative values indicating ecosystem C gain (Chapin et al., 2006)". The effect of climate change on NEE between 2035 and 2050 can be explained by more favorable conditions for tree growth (longer vegetation periods in the higher elevation parts of our mountainous study area) in combination with a $CO_2$ fertilization effect, relative to baseline climate conditions. We will combine Fig. 5, Fig. 6 and Fig. S14 as suggested by the referee in the revised version of the manuscript. We will also improve the text with regard to the interpretation of NEE in order to avoid confusing interpretations by future readers.

REFEREE #2: Supplementary materials The supplementary figures are excessive. Some could be merged into a single figure such as Fig. S11 and S12 that show the same variable (growing stock per species). Some are not even cited in the text such as Fig S13. Fig.S5 is not clear, why showing two sites in the fictitious landscape map with only on stand development below. Letters A to D are shown but not used in the explanation but the outcome of the spinup (letter D I guess?) is not highlighted.

AUTHORS: Figures S11 and S12 show the same variable, but provide different aspects of the simulation. While Fig. S11 compares the simulated with the observed species composition and growing stock in year 1999, Fig. S12 presents the temporal trajectory from 1905 to 2013 of the simulation only. The temporal trajectory cannot be provided for the observed data as there are no records available at annual resolution. Hence, by omitting Fig. S12 we would omit crucial complementary information. Fig. 13 is cited in the text in l. 387f. "At the same time total ecosystem carbon increased by 40.9% (Fig. S13)." Letters A to D have been explained in the supplement in l. 139 – 150 "For instance, the initial planting could plant trees according to the target species shares (A in Fig. S5). During the simulation the defined management steps are executed (e.g., thinnings, B, final cut C). Periodically, the state of the forest is evaluated against the available reference data. A basic evaluation compares, for instance, the growing stock and species shares emerging from the simulation with the respective reference state, and calculates a similarity score (e.g., Bray-Curtis index). When the deviation between the emerging state space from the simulations and the reference state are not satis-factorily, the STP for the next rotation can be altered. In the example in Fig. S5, the simulated share of spruce was lower than the spruce share in the reference state, in-dicating that spruce was likely favored by past management, either by planting spruce (C) or by favoring spruce via selective thinnings. This information is incorporated in the spin-up run, which henceforth uses a modified STP for the given stand and the next rotation (D)." In our opinion, the supplement figures all provide unique and com-plementary information, and are important to understand our approach and evaluate model behavior. As these figures will only appear in the online supplement and not the

main paper, we do not see a reason for reducing them, and thus withholding the details of our model evaluation efforts from the interested reader. Regarding Fig. S5 we agree with Referee #2 and will extend the figure caption to facilitate its interpretability.

REFEREE #2: Technical details P3 L49 'Keenan and others' instead of 'et al' the numeration of the supplementary materials is confusing with only one line of numbering for text sections and figures. There should be section S1, section S2, section S3, figure S1, figure S2, figure S3, figure S4

AUTHORS: We agree with the referees #1 and #2 that the structure of the supplement needs to be improved. We also thank the referee for his/her close view on the text, pointing out a mistake in the citation style. We will follow the referee's suggestion to differentiate between sections and figures, and correct the citation style where needed.

REFEREE #2: Crookston, N.L., Rehfeldt, G.E., Dixon, G.E., Weiskittel, A.R., 2010. Addressing climate change in the forest vegetation simulator to assess impacts on landscape forest dynamics. For. Ecol. Manag. 260, 1198–1211. Garcia-Gonzalo, J., Peltola, H., Gerendiain, A.Z., Kellomäki, S., 2007. Impacts of forest landscape structure and management on timber production and carbon stocks in the boreal forest ecosystem under changing climate. For. Ecol. Manag. 241, 243–257. Hurtt,G.C., Pacala, S.W., Moorcroft, P.R., Caspersen, J., Shevliakova, E., Houghton, R.A., Moore, B., 2002. Projecting the future of the US carbon sink. Proc. Natl. Acad. Sci. 99, 1389–1394. Karjalainen, T., Pussinen, A., Liski, J., Nabuurs, G.-J., Erhard, M., Eggers, T., Sonntag, M., Mohren, G.M.J., 2002. An approach towards an estimate of the impact of forest management and climate change on the European forest sector carbon budget: Germany as a case study. For. Ecol. Manag. 162, 87–103. Overpeck, J.T., Rind, D., Goldberg, R., 1990. Climate-induced changes in forest disturbance and vegetation. Nature 343, 51. Peng, C., Zhou, X., Zhao, S., Wang, X., Zhu, B., Piao, S., Fang, J., 2009. Quantifying the response of forest carbon balance to future climate change in Northeastern China: model validation and prediction. Glob. Planet. Change 66, 179–194. Seidl, R., Schelhaas, M.-J., Lexer, M.J., 2011. Unraveling the drivers of

intensifying forest disturbance regimes in Europe. Glob. Change Biol. 17, 2842–2852. Seidl, R., Schelhaas, M.-J., Rammer, W., Verkerk, P.J., 2014. Increasing forest disturbances in Europe and their impact on carbon storage. Nat. Clim. Change 4, 806.

Please also note the supplement to this comment: https://www.biogeosciences-discuss.net/bg-2018-145/bg-2018-145-RC2- supplement.pdf

AUTHORS: Thanks for providing the references and the pdf which has been more convenient to work with than the online version of the text.

References Grimm, V.: Pattern-Oriented Modeling of Agent-Based Complex Systems: Lessons from Ecology, Science (80-. )., 310(5750), 987–991, doi:10.1126/science.1116681, 2005. Mäkelä, A., Pulkkinen, M., Kolari, P., Lagergren, F., Berbigier, P., Lindroth, A., Loustau, D., Nikinmaa, E., Vesala, T. and Hari, P.: Developing an empirical model of stand GPP with the LUE approach: Analysis of eddy covariance data at five contrasting conifer sites in Europe, Glob. Chang. Biol., 14(1), 92–108, doi:10.1111/j.1365-2486.2007.01463.x, 2008. Marini, L., Ayres, M. P., Battisti, A. and Faccoli, M.: Climate affects severity and altitudinal distribution of outbreaks in an eruptive bark beetle, Clim. Change, 115(2), 327–341, doi:10.1007/s10584-012-0463-z, 2012. Overbeck, M. and Schmidt, M.: Modelling infestation risk of Norway spruce by Ips typographus (L.) in the Lower Saxon Harz Mountains (Germany), For. Ecol. Manage., 266, 115–125, doi:10.1016/j.foreco.2011.11.011, 2012. Pasztor, F., Matulla, C., Rammer, W. and Lexer, M. J.: Drivers of the bark beetle disturbance regime in Alpine forests in Austria, For. Ecol. Manage., 318, 349–358, doi:10.1016/j.foreco.2014.01.044, 2014. Pasztor, F., Matulla, C., Zuvela-Aloise, M., Rammer, W. and Lexer, M. J.: Developing predictive models of wind damage in Austrian forests, Ann. For. Sci., 72(3), 289–301, doi:10.1007/s13595-014-0386-0, 2015. Scheller, R. M., Spencer, W. D., Rustigian-Romsos, H., Syphard, A. D., Ward, B. C. and Strittholt, J. R.: Using stochastic simulation to evaluate competing risks of wildfires and fuels management on an isolated forest carnivore, Landsc. Ecol., 26(10), 1491–1504, doi:10.1007/s10980-011-9663-6, 2011. Schwaab, J., Bavay, M., Davin, E., Hagedorn,

F., Hüsler, F., Lehning, M., Schneebeli, M., Thürig, E. and Bebi, P.: Carbon storage versus albedo change: Radiative forcing of forest expansion in temperate mountainous regions of Switzerland, Biogeosciences, 12(2), 467–487, doi:10.5194/bg-12-467-2015, 2015. Seidl, R., Rammer, W., Scheller, R. M. and Spies, T. A.: An individual-based process model to simulate landscape-scale forest ecosystem dynamics, Ecol. Model., 231, 87–100, doi:10.1016/j.ecolmodel.2012.02.015, 2012a. Seidl, R., Spies, T. A., Rammer, W., Steel, E. A., Pabst, R. J. and Olsen, K.: Multi-scale drivers of spatial variation in old-growth forest carbon density disentangled with Lidar and an individual-based landscape model, Ecosystems, 15(8), 1321–1335, doi:10.1007/s10021-012-9587-2, 2012b. Seidl, R., Albrich, K., Thom, D. and Rammer, W.: Harnessing landscape heterogeneity for managing future disturbance risks in forest ecosystems, J. Environ. Manage., 209, 46–56, doi:10.1016/j.jenvman.2017.12.014, 2018. Senf, C. and Seidl, R.: Natural disturbances are spatially diverse but temporally synchronized across temperate forest landscapes in Europe, Glob. Chang. Biol., 24(3), 1201–1211, doi:10.1111/gcb.13897, 2018. Temperli, C., Bugmann, H. and Elkin, C.: Cross-scale interactions among bark beetles, climate change, and wind disturbances: A landscape modeling approach, Ecol. Monogr., 83(3), 383–402, doi:10.1890/12-1503.1, 2013. Thom, D., Seidl, R., Steyrer, G., Krehan, H. and Formayer, H.: Slow and fast drivers of the natural disturbance regime in Central European forest ecosystems, For. Ecol. Manage., 307, 293–302, doi:10.1016/j.foreco.2013.07.017, 2013. Thom, D., Rammer, W. and Seidl, R.: The impact of future forest dynamics on climate: interactive effects of changing vegetation and disturbance regimes, Ecol. Monogr., 87(4), 665–684, doi:10.1002/ecm.1272, 2017.

---

## Author Comment (AC3) · 29 May 2018

REFEREE #3: General comments : The research article named 'Disturbance legacies have a stronger effect on future carbon exchange than climate in a temperate forest landscape,' try to explore the effect of disturbance legacies and climate change in the projection of the forest carbon sequestration. In order to do that, they reconstruct a well documented historical scenario of an Austrian forest landscape with two disturbance events and one forest management shift. At the end of the paper, they encourage the scientific community to take into account the forest history when initializing the

forest state before running projections of the forest dynamic. This is a nice attempt to promote the integration of disturbances and abrupt mortality in model development. I really appreciate the quality of the work done by the simulation experiment and the past reconstruction forest state with the new method of spin-up. I am convinced that this paper can be published without deep changes in the structure and the content.

AUTHORS: We are grateful for the positive evaluation of our study.

REFEREE #3: However, five points need to be clarified: 1) The results of the simulation experiment show that past forest management (absence or presence) is the main factor to explain the divergence between simulations. But this finding is not central to the paper! Instead of that, the authors define forest management as a human disturbance (that is perfectly true) and merged natural and human disturbances in one general disturbance term. This merging leads to a misinterpretation of the title and the conclusion because, for most of the ecologist and the forest manager, disturbance legacies always refer to an extreme event legacy like storms, beetles outbreaks, fires or droughts. My advice is to explicitly divide interpretation of the result into the natural and the human disturbance. For example, the title will become: "Human disturbance/forest management/human activity legacies have a stronger effect on future carbon exchange than climate in a temperate forest landscape."

AUTHORS: We thank the reviewer for this important comment, and for the recommendations on how to improve our work further. This comment is congruent with one of the comments provided by Referee #2. As mentioned in the response to Referee #2, our attempt was to combine natural and human disturbances first in order to quantify the overall disturbance effect on carbon storage, and subsequently to disentangle the partial effects of natural and human disturbances. However, we understand the potential confusion this has been causing. In the revision we will clearly distinguish between management and natural disturbances throughout the study. We will also rephrase the title of the study into "Legacies of forest management have a stronger effect on future carbon exchange than climate and natural disturbances in a temperate forest

landscape".

REFEREE #3: 2) the authors need to be careful with the last statement of the title: "than climate in a temperate forest landscape" because the authors only realized simulations with a medium climate change scenario (A1B). The strongest climate change scenario like the RCP 8.5 is most likely to happen, and it will have a stronger impact. In addition, the authors forget to take into account the indirect effect of CC on forest growth via the increase of the frequencies and the intensities of the extreme events. This partly due to the setup of the simulation experiment where disturbances are forced and disconnected to the mortality module of iland. But this interaction can be simulated in iland because the authors already developed abrupt mortality module into this model.

AUTHORS: We agree with the referee that a more severe climate change scenario will likely alter the effect of climate change in our study. The exclusion of high intensity disturbance events after 2013 was necessary to exclude confounding effects from disturbance interactions with past disturbance events (i.e., spatio-temporal autocorrelation) in order to disentangle the partial effects of past disturbance and future climate change. In the revision, we will add this aspect explicitly to the discussion, highlighting possible impacts of a more severe climate change scenario on NEE. We will also mention the necessity to exclude high intensity disturbances in the methods section.

REFEREE #3: 3) The way the authors display the results of the simulation experiment is very confusing. The figure 5 for example which display the difference between reference NEE and alternative NEE, starts to diverge from 2013 and not from 1905. The simulations without management should not be far from other simulation in 2013?

AUTHORS: In order to derive the effect of management and natural disturbance legacies on the future trajectories of NEE we have defined the start point of the analysis after the second disturbance episode, i.e. in 2013. The figure thus presents the cumulative differences in carbon uptake or release resulting from this legacy effect of disturbance (comparing disturbed and undisturbed scenarios) as well as climate change

(comparing climate change and baseline climate) on the future NEE. The differences in total ecosystem carbon storage starting from year 1905 have been presented in Table 1. Based also on the comments of referee #2, we will omit Fig. 6 and combine Fig. 5 and Fig. S14. To avoid confusion, we will extend the figure caption, explaining more specifically how to interpret the newly added figure.

REFEREE #3: 4) In table 1, we can see a difference of about 40 tC ha between managed and unmanaged simulations. The strangest thing here is that in 2099 this difference disappears (compensation process?). This is interesting but the authors don't mention that in the discussion. Why? and why the figure 5 doesn't display that?

AUTHORS: As the referee points out correctly, Table 1 indicates that the differences in total ecosystem carbon storage between formerly disturbed and undisturbed scenarios become negligible by the year 2099. In other words, this shows that the legacy effect of past disturbances does not influence carbon storage beyond 2099. Fig. 5 corresponds to the output presented in Table 1 by showing that the cumulative carbon uptake levels off over time. Consequently, the differences in cumulative NEE in Fig. 5 at year 2099 correspond approximately to the differences in total ecosystem carbon storage in year 2013 between disturbed and undisturbed scenarios in Table 1 ($\sim$40 tC ha-1). The underlying reason for this compensatory effect is an increased growth (increased carbon uptake) by forests after disturbance. We will amend the discussion regarding the duration of the legacy effect of natural and human disturbances as well as the cause of the compensatory effect on NEE in the revised version of the manuscript.

REFEREE #3: 5) Did the two imposed disturbances have a different impact on the forest across simulations? If not, it means that the authors can't observe the legacy effect of one disturbance to another future one. It is maybe the reason why they don't observe a strong effect of natural disturbances. Due to this lake of interaction, the interesting questions like: - Can this forest have the capacity to absorb extreme events well enough to keep the same level of NEE if the intensity and the frequencies of natural disturbances will increase? Or - Are the forest management made between 1905 and

1997 is able to change disturbance impact on NEE in the future? cannot be tackled. It is a pity because it will strengthen the purpose of this paper.

AUTHORS: While we could not use the dynamic disturbance modules to mimic the first disturbance episode as we did not know its characteristics reasonably well to represent it in our process-based disturbance module (e.g., wind speed, wind direction), the second disturbance episode was in fact simulated dynamically, i.e., the simulation model produced different disturbance impacts on forests and carbon storage depending on the inclusion or exclusion of the first disturbance episode and forest management. The simulation design is explained to the reader in detail in l. 328 – 331: "From 1905 to 1923 management and natural disturbances were implemented in the simulation as recorded in the stand-level archival sources. After 1923, natural disturbances were simulated dynamically using the respective iLand disturbance modules." However, the aim of our study has not been to assess the effects of past natural and human disturbance on future disturbances. Instead, we excluded high mortality disturbance events in order to not confound the investigation of the legacy effects from past disturbances on NEE. Also in response to comments of other referees, we will provide a new analysis to investigate the contribution of the first disturbance episode and forest management on the second disturbance episode. In this way, we will derive some further insight into the legacy effects of natural and human disturbance on subsequent disturbance events.

---

## Author Response (AR1)

REFEREE #1:
General comments
The manuscript deals with the legacy effects of disturbances (both natural and anthropogenic), and of future climate change, on the C balance of the forest. It is a relevant topic and provides new input to the field. The manuscript is well-written and the work has been done thoroughly.

AUTHORS:
We thank the referee for his/her overall positive evaluation of our study.

REFEREE #1:
The first part of the study is an analysis of possible interactions between two past disturbance events. Although I can appreciate the work that has gone into digging out the old archives, my impression is that the analysis was more exploratory in nature, while writing it up, one reference (Schurman et al. 2018) was used as a quick excuse for a hypothesis and the discussion is more focussed to find references on temporal autocorrelations at different time scales. Perhaps part of the material in the discussion should be transferred to the introduction to provide a more solid hypothesis (like the references in line 442/443), or no hypothesis should be given at all and the patterns found should be discussed against other findings in literature. A weak point here is that there were only two events, and no autocorrelation analysis could be done at different time scales. Furthermore, I'm not always convinced by the arguments the authors bring up in comparing their results to other studies. For example, they state that they find a low probability for the same area to be affected by the two episodes (line 443), which is in contrast to a study that does find correlations between episodes but at very different timescales. I think there is only a contrast if both studies were at the same timescale, and if not, they cannot be compared. Similarly, they state that other studies did find correlations at the plot and stand scale (line 450), but the authors attribute their different finding to the fact that they work at the landscape scale. I do not see why this would yield so different results. If you check a sufficient number of stands and find correlations, I would expect the same would hold true for the landscape. If not, you would expect low correlations at the stand scale as well. Also, lines 457-466 pose some possible reasons why the two events were different. I think they should have enough material to check some of these alternative explanations, or should be able to obtain them with little effort (for example wind direction of both events). Overall, I suggest the authors re-think their hypothesis and discussion for this part of the analysis.

AUTHORS:
The referee makes an important point with regard to revising the hypotheses and the part of the discussion pertaining to the first part of our analysis. Given the lack of explicit data on past disturbance episodes, comparisons as the one undertaken here are rare, which makes embedding it in the literature challenging. Furthermore, as some important characteristics of the first disturbance episode remain unknown (e.g., exact wind speed, wind direction) some uncertainties about the causes of the difference between the two episodes will necessarily remain. Furthermore, we'd like to point out that an analysis of individual drivers of the Central European disturbance regime is beyond the scope of the current contribution, in fact the causes of natural disturbances have been investigated in detail already in prior studies (Marini et al., 2012; Overbeck and Schmidt, 2012; Pasztor et al., 2014, 2015; Thom et al., 2013). Nonetheless, we agree with Reviewer #1 that the correlation between the two disturbance events warrants further attention. Congruent with the suggestions of referee #1 and referee #2, we have added another analysis to investigate the contribution of past land use to the second disturbance episode (l. 364 – 372 in the manuscript version with track changes). Based on our factorial simulation design, we have now analyzed the effect of four different combinations of previous natural disturbances and management on the second disturbance episode in 320 simulations (those including the second disturbance episode). This analysis has revealed a high uncertainty about the relationship between both disturbance episodes, while past land use clearly increased disturbances on the landscape (l. 479 – 483). Following the referee's advice we have also reformulate our hypothesis and substantiated it with some of the material provided in the discussion section. Based on the results of the new analysis, we also have reformulated and extended the discussion in section 4.1.

REFEREE #1:
The second part of the study deals with an analysis of the future effect of human and natural disturbances, and future climate change. I think this part of the study is well described and the conclusions are valid. The authors give great care to initialise their model in 1905 using an innovative method, and to simulate the conditions until now, and then project their model into the future. They conclude that the past trajectory is very important to understand the future carbon dynamics. Usually, models would be initialised according to the current state of the forest, and carbon dynamics projected into the future. The current state of the forest would in most cases represent past events, and legacy effects are thus already present. I'm wondering if the 100-year simulation of the past really influences the results, and that this would be a recommended procedure for all models, or that the correct representation of current state and current management is sufficient to include these legacy effects. I could imagine the authors use their new initialisation procedure to represent the current state and compare future projections with and without the 100-year historic run. Perhaps this is too much to add to the current paper, but I would encourage the authors to give some indications on this issue. Are the current initialisation procedures sufficient to take care of past legacies or are longer historic runs needed?

AUTHORS:
We thank the referee for the positive evaluation of the second part of our study. The Reviewer is in fact correct in stating that usually legacy information is captured via the initialization of a model. This is not different in iLand, the model applied here. However, our point here is a slightly different one, namely: How different would the state of the forest (and hence the initialization of a simulation model) be if it would have had a different disturbance history? We thus quantify the structural effects of different past activities onto the state of the forest in 2013, and investigate how long these differences persist into the future, given everything else is equal. So the Reviewer is correct in assuming that if the initial conditions are known the legacies are adequately captured for modeling. However, in many cases the initial conditions of a forest landscape are incompletely known. This is for instance the case for the state of our landscape in 1905, for which we have information about species composition and growing stock, but not for other important variables (e.g., soil C pools, the spatial composition and configuration of stands). The legacy spin-up approach presented here was designed to address this very issue.

In the revision we have added some more explanation in section 2.4 in order to clearly distinguish between the different steps of our modeling approach, and to make explicitly clear what our contribution with the legacy spin-up is. In this regard, also the new arrangement of the supplement into sections helps to distinguish the legacy spin-up from subsequent simulations (see next comment).

REFEREE #1:
The ordering and numbering of the supplement is a bit strange. S2 and S3 are figures connected to text S1, S4 is text, while S5 and onwards are again figures. While reading the main text, the first reference is S4 while earlier supplementary material is referred to later. Perhaps the supplement could be ordered according to the appearance in the text, and a difference could be made between text and figures.

AUTHORS:
We agree that the enumeration of the supplement can be improved. This has also been suggested by referee #2.

In our revision, we have restructured the supplement into three sections and have provided all figure with a consecutive number. Sections and figures were numbered continuously throughout the text.

REFEREE #1:
Specific comments

In Figure 1 it would be helpful to add a small map to show where the study area is located within Austria.

AUTHORS:
We have complemented the figure with another panel showing a map of Austria and the location of the landscape.

REFEREE #1:
Line 152: does the model allow for build-up of beetle populations over the years?

AUTHORS:
Yes indeed, the process-based bark beetle module implemented in iLand is able to simulate the build-up of bark beetle populations over multiple years. Weather conditions affect the bark beetle population directly (e.g., the number of generations and sister broods per year, as well as winter survival rate). Furthermore, the vitality of trees and thus their defense capacity (simulated via the available non-structural carbohydrates) as well as the amount of windthrown trees (easily colonizable breeding material) influence beetle populations in the model. Seidl and Rammer (2017) found that iLand was well able to reproduce the 2$^{nd}$ bark beetle disturbance episode contained in our analysis here.
We have added this information in l. 188-196 in the revised version of the manuscript.

REFEREE #1:
Line 285: I assume the weather data was adapted to the elevation gradient in the study area somehow? If yes, could you add one sentence about it?

AUTHORS:
Indeed, elevation gradients are captured in the climate data used. The climate data from 1950 – 2099 were all statistically downscaled to a resolution of 100 x 100 m by means of quantile mapping. For the years 1905 – 1949, we had only temperature and precipitation from a nearby weather station. We thus drew a climate from the period 1950 – 2099 for each missing year by matching its temperature and precipitation data to that of the weather station record for 1905 – 1949.
We have extended the information about the downscaling approach of the climate data in l. 325-327.

REFEREE #1:
Line 356: Simulated species shares were compared against "independent" data for the year 1905. I think 1905 data were used to make the spin-ups for the model. If it is the same data, they are not independent. Or is it really another source? If so, please specify here.

AUTHORS:
We agree with the referee that the comparison of the simulation with the observed data is not entirely independent, as the observed data was used to guide the spin-up procedure.
We have changed the text accordingly in the revised version of the manuscript (l. 429-430).

REFEREE #1:
Line 410: Is "stock" perhaps better than "storage" here?

AUTHORS:
We have changed to "stock" in the revision.

REFEREE #1:
Line 487: You mention here that you only studied wind and bark beetles, while other agents may become more important in future. I think wildfire was included in your simulations as well. Moreover, you conclude that management was far more important than disturbances, i tihnk this needs to be highlighted here as well.

AUTHORS:

Our simulations included disturbances by wind and bark beetles only, as stated in l. 179-181 "As wind and bark beetles are of paramount importance for the past and future disturbance regimes of Central Europe's forests (Seidl et al., 2014a; Thom et al., 2013), we employed only these two process-based disturbance submodules in our simulations". Although it is correct that iLand is able to simulate disturbance from wildfire, we did not include wildfires here as they are not an important component of the disturbance regime in our study system.

With regard to management we agree on its importance, and have highlighted this more explicitly in the revision. Throughout the text we are now distinguishing between natural disturbances and land use, and, in particular, in l. 579-591 of the discussion we are pointing out the superior role of past land use as a driver of NEE.

**Anonymous Referee #2**

REFEREE #2:
This study depicts the past and future of a forest landscape in Austria. It aims at evaluating the respective weights of past natural disturbances, past human management, and future climate change on the forest capacity to sequester carbon. For this, the authors reconstructed the landscape history of the federal forest under study using historical data sources. This history is marked by a windstorm in 1905 followed by a bark beetle outbreak, technological evolution of management practices until 1997 when management is ceased, and a second wind and bark beetle event in 2007. The historical reconstruction results show that there is no correlation between the locations impacted by the first and the second natural disturbance events. In a second time, the authors designed a factorial simulations experiment in which the forest landscape under study undergoes all combinations of conditions : 1917 windstorm and bark beetle event or no, evolution of management practices between 1924 and 1997 or no management after 1924, 1997 windstorm and bark beetle event or no, four climate scenarios from 2013 to 2099. The simulations show that the net ecosystem exchange is dominated by past management found to explain 97.7%. The recovery from past management causes an increase in the future carbon storage. The authors find that by 2100 the effect of human and natural disturbances overcome the effect of climate change.
The object of this study is interesting and timely as the issue of the response of forests to climate change becomes more pressing. The case study is interesting due to its particular history including two large natural disturbance events and a ceasing of human management that allow the analysis of the legacy of management practices on a forest landscape. The simulation experiment is well designed and the model used (iLand) is appropriate to address the questions raised and introduced in a satisfactory way. However, the results and discussion section are somewhat superficial and do miss some important points. Also, the way the study is presented is often confusing or misleading and impairs the comprehension and interpretation of the results. The display items as well as the presentation of the results should be reconsidered to enhance the impact of the work presented.

AUTHORS:
We thank the referee for his/her interest in our study and the very thoughtful review with valuable comments to help us improve our manuscript.
In the revision, we have particularly focused on dissolving the confusing interpretation of results that were highlighted by the reviewer. See our responses below on how we achieved this.

REFEREE #2:
Detailed comments
Terminology : " disturbance " My main concern is about the use of the word disturbance all along the article, from the title on. The use of this term disturbance is misleading. Usually disturbance refers to natural disturbance (Overpeck et al., 1990; Seidl et al. 2014, 2011). In the present manuscript, it is sometimes used to refer to natural disturbances only (p4 L73 or L395) and sometimes to refer to natural + anthropogenic. It seems that the authors are aware of the confusion this creates, because most times they explicit that disturbances is natural+anthropogenic (ex : p5L86). Aggregating two very different processes such as management and natural disturbances, on top of being very confusing for the reader, impedes the discussion of one very important result which is the extreme dominance of the effects of management compared to natural disturbances on carbon sequestration of forests. To this regard even the title of the article is misleading or even incorrect since it is not the legacy of the natural disturbance events (explaining only 2,8%) but of past management that has a stronger legacy effect than climate change. The manuscript should be revised to account explicitly for this distinction in the processes analyzed which is obvious in the results.

AUTHORS:
We thank the Reviewer for pointing this out. Our idea in the initial submission was to first combine natural and human disturbances to quantify the overall disturbance effect on carbon storage, and subsequently disentangle the partial effects of natural and human disturbances. As two of the three referees (referee #2 and referee #3) found the combination of natural and human disturbances into the overall disturbance effect confusing and problematic, we concede that this idea had to be revised.
In the revision we now clearly distinguish between land use and natural disturbance throughout our study. We have also rephrased the title of the study into "Legacies of past land use have a stronger effect on forest carbon exchange than future climate change in a temperate forest landscape".

REFEREE #2:
Methods
In the description of the simulation experiment it is noted that each scenario is replicated 20 times (p15 L 347) ? The rationale for this should be explained. What changes between the replicates ? Is there a stochastic component in the model ?

AUTHORS:
iLand is a process-based model including fully-dynamic submodules for natural disturbances and forest management, and each of these components contain stochasticity (e.g., the spread of an individual bark beetle cohort from an infested tree is determined by drawing from a distribution of empirically determined dispersal distances, with spread direction drawn randomly between 0° and 360°). To account for this stochasticity, we have replicated every simulation 20 times. This particular number has been proven to be a good middle ground between determining robust results and keeping simulation times reasonable in previous applications of the model (e.g., Seidl et al., 2018; Thom et al., 2017).
We have now provided the rationale of the replicates more explicitly in l. 417-420 (see manuscript version with track changes).

REFEREE #2:
L212: the sentence describing the 1905 age distribution seems a bit far-reaching from fig S8 as the bimodal distribution is not obvious, and the statement is very qualitative.

AUTHORS:
We agree with the referee and have changed the text accordingly (l. 253-255).

REFEREE #2:
Results and discussion
The manuscript seems very unbalanced with 13,5 pages of intro and methods (both well written and with relevant content) and only 5,5 pages of results and discussion (2,5 and 3 respectively). As reflected by these numbers, the results and discussion sections are sometimes shallow compared to the information presented and the very large number of display items included both in the main text and the supplementary materials (8 and 12 respectively).

AUTHORS:
We only partly agree with the referee on this point. We feel that it is important to include an extensive methods section in highly complex and computational extensive studies in order to ensure the highest possible degree of reproducibility (see also Scheller et al., 2011; Schwaab et al., 2015; Temperli et al., 2013). As 5 of the 8 figures as well as both tables are anchored in the results section, the manuscript is overall less imbalanced as it may seem based on text pages only. Moreover, besides the 5.5 pages for results and discussion, there is another page of text making up the conclusion section, which should be considered as well. With regard to the supplement we feel that the extensive additional material presented here helps the reader to understand our study and provides additional context on the validation and applicability of the methods used in our study (while not further clogging the main text).
Nonetheless, we have further strengthened the results and discussion sections in the revised version of the manuscript through an additional analysis of the impacts of management and the first disturbance episode on the second disturbance episode (see our response below). Additionally, we have improved the clarity of information provided in the results and discussion sections based on reviewer comments. However, we refrained from omitting parts of the methods (which the referee agrees are relevant to understand the study) or prolong the results and discussion sections extensively (as our manuscript is already fairly long).

REFEREE #2:
Some missing information : - 3.1 Performance of the reconstruction of past events:L377 " a good match" with reference to three supplementary figures, L379 " well able " with reference to one supplementary figure, L381 "small overestimation", L382 "corresponded well" with reference to one supplementary figure etc. all results from section 3.1 are qualitative and based on supplementary figures. An effort should be made to quantify the quality of the reconstruction and to present it in a concise manner in one display item, that, if judged crucial for the validity of the results should be presented in the main text.

AUTHORS:
We understand the desire of Referee #2 for a single, concise evaluation result. However, we here follow a patter-oriented modeling approach (Grimm et al., 2005), which means that a variety of very different indicators are considered in order to evaluate the model's ability to reproduce the empirically derived historic data (i.e., tree species composition in 1905 and 1999, management, natural disturbances). In our opinion these cannot be combined into a single number/ figure, as such a combination may hide important information regarding model performance (e.g., the model could be doing very well wrt one indicator while performing poorly wrt a second one, which would give on average moderate performance; if the poorly captured indicator is, however, of particular importance for the study, this information would be lost in such an aggregate evaluation). After careful consideration of the Referees comment we thus have decided to retain the multidimensional nature of our evaluation.

In the revision we have explained this in more detail and provided our rationale for this approach for the reader in l. 350-355.

REFEREE #2:
- 3.2 Temporal interaction of disturbance events: the autocorrelation between natural disturbance events is described and found very low. No link is analyzed between disturbance events and management: is there a correlation between stands affected by natural disturbance and species? And age? And density?

AUTHORS:
We thank the Referee for bringing up the issue of management in this context. In fact the possibility of a connection between management and the second disturbance episode has also been pointed out by referee #1, and we agree that this is an important issue here. We thus have now added an additional analysis in this regard to our manuscript.

Following the advice of referees #1 and #2, we have investigated the contribution of land use on the second disturbance episode in our revision. In particular, we have analyzed the effect of all 4 potential combinations of past natural disturbance and land use on the second disturbance episode in 320 simulations (i.e., those including the second disturbance episode). This additional analysis has helped us to investigate legacy effects of past land use and natural disturbance on subsequent disturbances. In particular, we found only a moderate non-significant effect of the first disturbance episode on the second disturbance episode, while past land use had a strong and significant impact on the second disturbance episode. These new results are presented in l. 479-483, and are further discussed in l. 557-567.

REFEREE #2:
- 4.1 The discussion of the lack of autocorrelation between both natural disturbance events and the link to previously published literature is not always clear. For example, the authors state that their hypothesis was that older stands are more prone to wind and bark beetle damages (L442) and link this statement to the low probability of a same area to be affected twice. The fact that a stand is affected by a disturbance does not make it older hence more susceptible to a second disturbance. Several hypotheses are formulated to explain the lack of autocorrelation between both episodes as found in other studies, but none is backed by data so that the discussion is not convincing. One hypothesis is that the longer and larger temporal and spatial scales analyzed here weaken the link found in smaller scale studies. I do not see why stands being more prone would not show up at the landscape scale. Similarly, the hypothesis of a dampening effect of a previous disturbance due to the resulting heterogeneity should be backed by minimal tests on the age and species structures of the affected and non affected stands. As well, the suggestion as to the difference in wind directions of both events needs to be investigated. In summary, an analysis of the characteristics of the stands affected by both natural disturbance events would enlighten this part.

AUTHORS:
As highlighted by referees #1 and #2 we agree that this part of the discussion needed to be revised. We have tried to find other studies investigating the spatial autocorrelation of two consecutive major disturbance episode, but spatio-temporal autocorrelation of disturbances has been usually either described over very limited time frames (e.g., Pasztor et al., 2014) or the spatial resolution for the comparison of disturbances over longer time frames has been very coarse (e.g., Senf and Seidl, 2018). In this regard, our analysis constitutes a novel contribution, improving our understanding of disturbance dynamics over extended temporal scales. Although we have spatially explicit disturbance data for both events, we cannot conduct a process-based analysis at the level of individual drivers. The reason is that we do not know all the characteristics of the wind event of the 1917-1923 disturbance episode (e.g., wind direction and wind speeds) as these have not been faithfully documented. Moreover, we feel that the analysis of disturbance drivers is beyond the scope of the current contribution, as this has been investigated in more detail in other studies in Central European ecosystems (e.g., Marini et al., 2012; Overbeck and Schmidt, 2012; Pasztor et al., 2014, 2015; Thom et al., 2013). Nonetheless, we have improved the analysis of how past legacies have affected recent disturbances in the revised version of the manuscript. As mentioned above we have added a new analysis investigating the contribution of the first disturbance episode and forest management on the second disturbance episode. This analysis has served to substantiate our finding of a weak contribution of one disturbance episode on the other, and provides more insights into the effect of forest management on the Central European disturbance regime. Based on these results we have also improved the discussion in section 4.1 of the revised manuscript. Moreover, following the Reviewer's advice, we omitted the comparison of our results with other studies investigating autocorrelation between natural disturbance events at different temporal and spatial scales.

REFEREE #2:
-4.2 disturbance legacies on future C uptake The authors argue that other studies of effect of climate change on carbon sink do not explicitly consider the legacy of past events. It is a bit surprising as past events' legacy in embedded in the initial conditions. The legacy spinup method derived here is interesting and relevant but should be placed in the context of alternative methods to describe forest initial conditions, see for example (Crookston et al., 2010; Garcia-Gonzalo et al., 2007; Hurtt et al., 2002; Karjalainen et al., 2002; Peng et al., 2009). The novelty of this study does not seem to be the inclusion of the disturbances' legacy but their quantification so this section should be rephrased. Several sentences are not backed by any reference and should be justified and developed. For example on L484, the sentence stating that these results may not hold for longer time frames, on L499 the sentence interepreting the simulation results as a change in forest types.

AUTHORS:
We agree with the Reviewer that the legacy effects are indeed embedded in the initial conditions, if the initialization is based on a comprehensive set of empirical data. It is also correct that the quantification of the legacy effect is the actual novel contribution of our study (see also our response to a similar comment of Referee #1).

We have rephrased this section to indicate that only few studies have quantified the legacy of past natural disturbances and forests management to date. However, we feel that further discussion about the legacy spin-up would decrease the focus of this section, as the legacy spin-up refers to the landscape history before 1905, while this section addresses the legacies of the disturbance episodes in 1917-1923 and 2007-2013 as well as land use between 1905-1997 on future trajectories. Instead, we have extended the supplement with a discussion of empirical initialization approaches (l. 110-122 in the version with track changes), followed by a comparison of traditional spin-up approaches with the legacy spin-up developed for this study (l. 123-151).

REFEREE #2:
- effect of climate change It is not explained in many details what response of forest growth to climate change is simulated by iLand (with respect to species or altitude for example). The results shown here on the comparison of climate change and management are highly related to the processes included in the modeling exercise as correctly stated in L501-507 and would deserve a more in-depth explanation. A discussion section on the simulated response of forest growth to climate change only would help put the results in perspective.

AUTHORS:
We agree that this is important information for readers in order to understand the results presented here. iLand considers both direct and indirect vegetation responses to climate change. For instance, temperature increases directly affect processes such as leaf phenology and the length of vegetation period, the efficiency of photosynthesis (modeled using a state acclimation approach following Mäkelä et al. 2008), and the availability of water in the soil (via altered evapotranspiration rates). Similarly, rising $CO_2$ levels directly affect net primary production via $CO_2$ fertilization. Thus, climate change might affect growth of one species differently than that of another species (direct effect), leading to a change in forest competition and structure (indirect effect).

We have provided more details of the climate change effects on forest vegetation in iLand in l. 167-174. We are also explicitly referring the reader to the more technical iLand papers describing this issue in detail (Seidl et al., 2012b, 2012a; Thom et al., 2017).

REFEREE #2:
Display items
Some display items do not help the understanding of the text, are redundant, or at the contrary lack information, and so should be rethought as material that supports the claim made in the text. Fig2 aims at summarizing the events included in the historical reconstruction of the forest landscape. Its design is more appropriate for a slideshow than a written article. Fig3 illustrates how the events shown in fig2 are included in the simulation experiments. Its design is confusing, especially with the 'n' that is cumulated from left to right (it takes some time to understand this) and that attempts at expliciting the factorial combination of the events simulated. These 2 figures could be condensed into a single display item where only the information relevant to the study would be presented. For example a table structured as below:
Period / Scenarios' options / details
1905-1924 / disturbed / storm+bark beetle+
...
/ undisturbed /
1924-1997 / managed / Technological improvements
/ unmanaged / Forest left to grow
1997-2013 / disturbed /
/ undisturbed /
2013-2099 / Climate scenario 1 /
/ Climate scenario 2 /
/ Climate scenario 3 /
/ Climate scenario 4 /

AUTHORS:
We have evaluated different options to combine both figures, but haven't found a satisfying solution. As mentioned by the Referee, Figures 2 and 3 are highlighting two different aspects of the study: Figure 2 represents the history of events on the landscape while Figure 3 shows the simulation design.
In order to provide the reader with a visual impression of the historic events relevant for this study, we decided to retain Fig. 2 instead of converting it into a table. Following the reviewer's advice, we have omitted Fig. 3 to avoid confusion and redundancy, and instead elucidated the simulation design in more detail in the text (l. 414-417).

REFEREE #2:
Other problematic display items are Fig5, Fig6 and FigS14. These three figures are redundant and should be combined into a single figure that shows the time evolution of NEE attributed to climate, event1, event2, and management. Please explain 'cumulative NEE'. From fig5, since the climate driven cumulative NEE decreases it means that the forest becomes a source of carbon between 2035 and 2050? This pattern should be discussed (see comment on 'effect of climate change' ).

AUTHORS:
The referee is right that there is some redundancy between these figures, as the endpoints in Fig. 5 and Fig. S14 reflect the effect size in Fig. 6. However, the interpretation of NEE by the Referee is not correct here. As NEE = -NEP a decrease in NEE means an increase of carbon in terrestrial ecosystems, i.e., between 2035 and 2050 there is an uptake of carbon by forests under climate change. We have provided a definition of NEE in l. 363f.: "NEE denotes the net C flux from the ecosystem to the atmosphere, with negative values indicating ecosystem C gain (Chapin et al., 2006)". The effect of climate change on NEE between 2035 and 2050 can be explained by more favorable conditions for tree growth (longer vegetation periods in the higher elevation parts of our mountainous study area) in combination with a $CO_2$ fertilization effect, relative to baseline climate conditions.

We have combined Fig. 5, Fig. 6 and Fig. S14 as suggested by the referee in the revised version of the manuscript. We have also improved the text in l. 116-119 and in the figure caption with regard to the interpretation of NEE in order to avoid confusing interpretations by future readers.

REFEREE #2:
Supplementary materials
The supplementary figures are excessive. Some could be merged into a single figure such as Fig. S11 and S12 that show the same variable (growing stock per species). Some are not even cited in the text such as Fig S13. Fig.S5 is not clear, why showing two sites in the fictitious landscape map with only on stand development below. Letters A to D are shown but not used in the explanation but the outcome of the spinup (letter D I guess?) is not highlighted.

AUTHORS:
Figures S11 and S12 show the same variable, but provide different aspects of the simulation. While Fig. S11 compares the simulated with the observed species composition and growing stock in year 1999, Fig. S12 presents the temporal trajectory from 1905 to 2013 of the simulation only. The temporal trajectory cannot be provided for the observed data as there are no records available at annual resolution. Hence, by omitting Fig. S12 we would omit crucial complementary information. Fig. S13 (now Fig. S11) was cited in the text in l. 462. "At the same time total ecosystem carbon increased by 40.9% (Fig. S11)." Letters A to D have been explained in the supplement in l. 162 – 173 "For instance, the initial planting could plant trees according to the target species shares (A in Fig. S5). During the simulation the defined management steps are executed (e.g., thinnings, B, final cut C). Periodically, the state of the forest is evaluated against the available reference data. A basic evaluation compares, for instance, the growing stock and species shares emerging from the simulation with the respective reference state, and calculates a similarity score (e.g., Bray-Curtis index). When the deviation between the emerging state space from the simulations and the reference state are not satisfactorily, the STP for the next rotation can be altered. In the example in Fig. S5, the simulated share of spruce was lower than the spruce share in the reference state, indicating that spruce was likely favored by past management, either by planting spruce (C) or by favoring spruce via selective thinnings. This information is incorporated in the spin-up run, which henceforth uses a modified STP for the given stand and the next rotation (D)."
In our opinion, the supplement figures all provide unique and complementary information, and are important to understand our approach and evaluate model behavior. As these figures will only appear in the online supplement and not the main paper, we do not see a reason for reducing them, and thus withholding the details of our model evaluation efforts from interested readers. Regarding Fig. S5 we agree with Referee #2 and have extended the figure caption to facilitate its interpretability.

REFEREE #2:
Technical details
P3 L49 'Keenan and others' instead of 'et al' the numeration of the supplementary materials is confusing with only one line of numbering for text sections and figures. There should be section S1, section S2, section S3, figure S1, figure S2, figure S3, figure S4

AUTHORS:
We agree with the referees #1 and #2 that the structure of the supplement needed to be improved. We also thank the referee for his/her close view on the text, pointing out a mistake in the citation style.
We have followed the referee's suggestion to differentiate between sections and figures, and corrected the citation style where necessary.

REFEREE #2:

Crookston, N.L., Rehfeldt, G.E., Dixon, G.E., Weiskittel, A.R., 2010. Addressing climate change in the forest vegetation simulator to assess impacts on landscape forest dynamics. For. Ecol. Manag. 260, 1198–1211.

Garcia-Gonzalo, J., Peltola, H., Gerendiain, A.Z., Kellomäki, S., 2007. Impacts of forest landscape structure and management on timber production and carbon stocks in the boreal forest ecosystem under changing climate. For. Ecol. Manag. 241, 243–257.

Hurtt,G.C., Pacala, S.W., Moorcroft, P.R., Caspersen, J., Shevliakova, E., Houghton, R.A., Moore, B., 2002. Projecting the future of the US carbon sink. Proc. Natl. Acad. Sci. 99, 1389–1394.

Karjalainen, T., Pussinen, A., Liski, J., Nabuurs, G.-J., Erhard, M., Eggers, T., Sonntag, M., Mohren, G.M.J., 2002. An approach towards an estimate of the impact of forest management and climate change on the European forest sector carbon budget: Germany as a case study. For. Ecol. Manag. 162, 87–103.

Overpeck, J.T., Rind, D., Goldberg, R., 1990. Climate-induced changes in forest disturbance and vegetation. Nature 343, 51. Peng, C., Zhou, X., Zhao, S., Wang, X., Zhu, B., Piao, S., Fang, J., 2009. Quantifying the response of forest carbon balance to future climate change in Northeastern China: model validation and prediction. Glob. Planet. Change 66, 179–194.

Seidl, R., Schelhaas, M.-J., Lexer, M.J., 2011. Unraveling the drivers of intensifying forest disturbance regimes in Europe. Glob. Change Biol. 17, 2842–2852.

Seidl, R., Schelhaas, M.-J., Rammer, W., Verkerk, P.J., 2014. Increasing forest disturbances in Europe and their impact on carbon storage. Nat. Clim. Change 4, 806.

Please also note the supplement to this comment:
https://www.biogeosciences-discuss.net/bg-2018-145/bg-2018-145-RC2-supplement.pdf

AUTHORS:
Thanks for providing the references and the pdf which has been more convenient to work with than the online version of the text.

AUTHORS:
While we could not use the dynamic disturbance modules to mimic the first disturbance episode as we did not know its characteristics reasonably well to represent it in our process-based disturbance module (e.g., wind speed, wind direction), the second disturbance episode was in fact simulated dynamically, i.e., the simulation model produced different disturbance impacts on forests and carbon storage depending on the inclusion or exclusion of the first disturbance episode and forest management. The simulation design is explained to the reader in detail in l. 395 – 398: "From 1905 to 1923 management and natural disturbances were implemented in the simulation as recorded in the stand-level archival sources. After 1923, natural disturbances were simulated dynamically using the respective iLand disturbance modules." However, the aim of our study has not been to assess the effects of past natural and human disturbance on future disturbances. Instead, we excluded high mortality disturbance events in order to not confound the investigation of the legacy effects from past disturbances on NEE.
Also in response to comments of other referees, we have added a new analysis to investigate the contribution of the first disturbance episode and forest management on the second disturbance episode (l. 364 – 372). This analysis has revealed a high uncertainty about the relationship between both disturbance episodes, while past land use clearly increased disturbances on the landscape (l. 479 – 483) We further discuss these new results in l. 557 – 567.

**Legacies of past land use have a stronger effect on forest carbon exchange**

**than future climate change in a temperate forest landscape**

**Running head**: " Land use legacies determine C exchange"

Dominik Thom[* 1,2], Werner Rammer[1], Rita Garstenauer[3,4], Rupert Seidl[1]

[1] Institute of Silviculture, Department of Forest- and Soil Sciences, University of Natural

Resources and Life Sciences (BOKU) Vienna, Peter-Jordan-Straße 82, 1190 Vienna, Austria

[2] Rubenstein School of Environment and Natural Resources, University of Vermont, 308i Aiken

Center, Burlington, VT 05405, USA. Tel: +1 802 557 8221. Fax: +1 802 656 2623. Email:

dominik.thom@uvm.edu

[3] Institute of Social Ecology, Department of Economics and Social Sciences, University of

Natural Resources and Life Sciences (BOKU) Vienna, Peter-Jordan-Straße 82, 1190

Vienna, Austria

[*] Corresponding author

**Abstract**

Forest ecosystems play an important role in the global climate system, and are thus intensively discussed in the context of climate change mitigation. Over the past decades temperate forests were a carbon (C) sink to the atmosphere. However, it remains unclear to which degree this C uptake is driven by a recovery from past land use and natural disturbances or ongoing climate change, inducing high uncertainty regarding the future temperate forest C sink. Here our objectives were (i) to investigate legacies within the natural disturbance regime by empirically analyzing two disturbance episodes affecting the same landscape 90 years apart, and (ii) to unravel the effects of past land use and natural disturbances  as well as future climate on 21st century forest C uptake by means of simulation modelling. We collected historical data from archives to reconstruct the vegetation and disturbance history of a forest landscape in the Austrian Alps from 1905 to 2013. The effects of  legacies and  climate were  disentangled by individually controlling for past land use, natural disturbances, and future scenarios of climate change in a factorial simulation study. We found only moderate spatial overlap between two episodes of wind and bark beetle disturbance affecting the landscape in the early 20th and 21st century, respectively. Our simulations revealed a high uncertainty about the relationship between the two disturbance episodes, whereas past land use clearly increased the impact of the second disturbance episode on the landscape. The future forest C sink was strongly driven by the cessation of historic land use, while climate change reduced forest C uptake. Compared to land use change the two past episodes of natural disturbance had only marginal effects on the future carbon cycle.

We conclude that neglecting  legacies can substantially bias assessments of future forest dynamics.

**Key words**: bark beetles, climate change, forest history, forest management, Kalkalpen

National Park, legacy effects, net ecosystem exchange, wind

**50 Copyright statement**

The authors agree to the copyright statement as described at https://www.biogeosciences.net/about/licence_and_copyright.html.

[revised manuscript text omitted]
. In forest landscape models, tThe initialization of the state of the ecosystems in forest landscape models accounts for legacies of past managementland use and disturbance legacies, if the data is based on empirically derived records. TherebyHowever, the level of detail required for theinformation provided upon initialization differs considerably between models (e.g., Garcia-Gonzalo et al., 2007; Schumacher and Bugmann, 2006; Thom et al., 2017) and is crucially determined by model structure. For instance, while forest structurale information plays only a minor role in pixelcell-based simulation models (Scheller et al., 2007), individual-based models requireretain information about tree dimensions, canopy heights, gaps, regeneration etc. (Seidl et al., 2012). Yet, detailed information about forest history ecosystem attributes for initializing simulation models is oftentimes not available (e.g., the spatial patterns of past disturbances or). initial belowgroundsoil carbon stocks). This is important as Uuncertainties in initialization can have substantial influence on the simulated trajectories (Temperli et al. 2013).

Using models enables the simulation of past forest development, including past management and disturbances, in the form of a spin-up run. Models can thus help to create realistic and quantitative past and current states of forests. In a conventional spin-up, the model is run for an extended period of time under past forcing, and a snapshot of the simulated state is taken– after reaching a predefined stopping criterion (e.g., elapsed time, variation in certain C pools) – as the starting point for scenario analyses (Thornton and Rosenbloom 2005). This results in meaningful estimates regarding important ecosystem properties, and a system state that is consistent with the internal model logic. However, thus derived ecosystem states often do not correspond well with the information available from past and current observations. For instance, a stand that was recently disturbed in reality could be initialized in a late-seral stage from a spin-up. This lack of structural realism strongly limits the utility of a traditional spin-up approach for initializing models for future projections. Factors such as the spatial distribution of age cohorts on the landscape have important implications for the future ecosystem dynamics, e.g., in the context of future susceptibility to disturbances. Therefore, we have developed a new spin-up approach, termed legacy spin-up, aiming to assimilate available data on the ecosystem state at a given point in time into the spin-up procedure, in order to improve the correspondence of the model state derived from spin-up with the observed state of the system.

[revised manuscript text omitted]

Thornton P, Rosenbloom NA. 2005. Ecosystem model spin-up: Estimating steady state conditions in a coupled terrestrial carbon and nitrogen cycle mode. Ecol Modell 189:25–

48.

## Section S3: Model evaluation and forest development after 1905

[Figure]

Fig. S79: Growing stock (timber volume over bark) harvested in the periods (a) 1924 – 1952, (b) 1956 – 1973, and (c) 1974 – 1983, as reconstructed from archival sources (observed) and simulated with iLand. Simulation data are for the baseline scenario, i.e. assuming historic natural disturbances and management regimes.

[Figure]

Fig. S8̶1̶0̶: Observed and simulated growing stock disturbed during the second disturbance episode (2007 – 2013). Observed values were derived from disturbance inventories of

Kalkalpen National Park, whereas simulated values are for the baseline scenario (i.e., assuming historic natural disturbances and management regimes.

[Figure]

Fig. S911: Observed and simulated growing stock by tree species in the year 1999. Observations are from forest management and planning data  of the Austrian Federal Forests, whereas simulated data are for the baseline scenario (i.e., assuming historic natural disturbance and management regimes).

[Figure]

Fig. S10: Growing stock by tree species over time, reconstructed by means of simulation modeling. Data are for the baseline scenario (i.e., assuming historic natural disturbance and management regimes).

[Figure]

Fig. S11: Carbon storage per compartment, reconstructed by means of simulation modeling. Data are for the baseline scenario (i.e., assuming historic natural disturbance and management regimes).

[Figure]

Fig. S14: Mean cumulative change in NEE induced by disturbance, distinguishing the effects of management from that of the first and second episode of natural disturbances. Shaded areas denote the standard deviation (SD) in NEE over the respective scenarios. Please note that panels are scaled individually.

---

## Author Response (AR2)

ASSOCIATE EDITOR:

Dear Authors,

In the light of the referee reports, I would like to invite you to prepare a revised manuscript that addresses the concerns of both referees (or explains in a cover letter why the comments were not taken into account). I share the concern of referee #2 with respect to the use of subjective statements such as "good match", "well able", etc. These statements should be made objective.

AUTHORS:

We are grateful for the positive evaluation of our manuscript by the editor and reviewers. In the revision, we have addressed the remaining comments. In particular, we have quantified our previously qualitative statements throughout the manuscript.

REFEREE #2:

I appreciate the efforts that the authors made to make the manuscript clearer and to elucidate some confusing points. The revised manuscript addressed many of my concerns and results more persuasive. The additional analyses bring new and interesting information to the study. However, I am not convinced by some of the authors' edits.

AUTHORS:

We thank the reviewer for his/her positive evaluation of our manuscript. We have addressed all remaining remarks in the revised version of the manuscript.

REFEREE #2:

1-I understand that the use of a pattern-oriented modeling approach (Grimm et al., 2005) is used to argue for a qualitative and multi-indicator description of model results. Maybe my comment was not clear. It is not a question of finding a single concise value, but simply to give the values that lead to the interpretation of the results that is currently presented in some parts of the results section. Statements such as « well able», «good match » do not describe the results but your interpretation of the results. Such statements would be acceptable in the discussion section but the objective description of the results is an important part of any scientific work, which I find sometimes missing still in the revised manuscript and relates to my prior comment on the shallowness of the results and discussion section. Statements such as « Good match » mean different things for different people which is not what is expected in the presentation of scientific results that should state objectively state what was found and allow the reader to judge for him/herself if the match is good/sufficient. The interpretation can then be discussed later on. I notice that this is actually done well in most part of the results section.

In brief, I do not agree that using a variety of indicators prevents the quantified description of the observed patterns. In particular, this argument hardly applies to the displayed results which are shown ad one-variable figures so it is merely a question of writing out in text what is displayed in the figure. In this case, stating in the text the numbers shown in the figure does not undermine the multi-criteria facet of your analysis.

AUTHORS:

We thank the reviewer for the clarifications about his/her concerns regarding the subjective statements. We agree that the quantification of these statements improves the objectivity of our study. Hence we have quantified all these following the reviewer's advice.

REFEREE #2:

In particular, please address these statements :

-       « The results from the legacy spin-up revealed a good match with the   species composition and growing stock expected from the historic records for the year 1905   (see Section S2, including Fig. S5, Fig. S6).    » Fig S5 shows the variable species share in simlated versus reference state, Fig S6 shows the growing stock for simulated versus reference state. A more rigorous presentation of the results would read something in the line of «The species composition from the legacy spinup diverged by XX%, YY%, ZZ% from the reference state for species X, Y and Z respectively while the simulated growing stock average was 207, 4% lower than the reference state ».

AUTHORS:

We addressed this comment in l. 415-417 of the revised version of the manuscript (with track changes).

REFEREE #2:

-       « ABE was well able (fig S7) » Here Fig S7 again only shows one variable, hampering the multi-indicator justification for not expliciting the results.

Response: We addressed this comment in l. 419-421 of the revised version of the manuscript.

REFEREE #2:

- "small overestimation", refers only to simulated harvest. Please quantify the variable described.

AUTHORS:

We addressed this comment in l. 422-423 of the revised version of the manuscript.

REFEREE #2:

- "corresponded well" referes to the disturbed growing stock only, please give in the text the numbers shown in the figure.

AUTHORS:

We addressed this comment in l. 422-423 of the revised version of the manuscript.

REFEREE #2:

- « Climate change weakened the carbon sink strength …» by how much ? by reducing the growing stock by XX%, YY%, ZZ% etc for species X,Y and Z resp. ?

AUTHORS:

We addressed this comment in l. 470-472 of the revised version of the manuscript.

REFEREE #2:

- « climate change effects on NEE were more variable compared to disturbance legacy effects, with increasing uncertainty over time as a result of differences in climate scenarios ». Give standard deviation to average ratio for each factor.

AUTHORS:

We addressed this comment in l. 472-477 of the revised version of the manuscript.

REFEREE #2:

- Figure 4 that results from the merging of Fig 5, 6, S14 is an improvement and make the results easier to grasp. However, the use of this figure in the text is not only to describe the evolution of each factor but to support the main result of the study stated in the title : the largest importance of land use effects over disturbances. This result is not easy to read from the different scales on the current figure. Please consider adding to Fig. 4 a panel with all 4 factors displayed on the same scale which would make obvious the main result.

AUTHORS:

The reviewer makes a good point that it would be easier to access the main finding of our study (a stronger effect of past land use on future NEE compared to climate change and natural disturbance), if the four drivers of NEE change were illustrated in one panel. Following the reviewer's advice, we have prepared another panel showing the effects of the four drivers of NEE at the same scale, and retained the previous figure showing the individual effects in more detail (based on a previous comment by one of the reviewers).

REFEREE #2:

− « Based on our simulations we found only a moderate positive effect of the first disturbance episode on the volume disturbed during the   second episode (+8,181 m , p=0.401). In contrast, land use had a considerable impact on the   second disturbance episode. On average, land use increased the volume disturbed by +28.927   m (p<0.001).   »

This result of the new analyses is important but not clear. Which display items does it refer to and why is the unit « m » ?

AUTHORS:

We do not know why the reviewer found the unit in m instead of m³. We confirmed the correct unit in the previous version of the manuscript by downloading it from the Copernicus submission system. Based on previous reviewer suggestions to reduce the number of display items, and the consideration that this finding will neither make a good display item nor an accurate table, we have not extended our manuscript with another figure.

REFEREE #2:

Details :

L425 of track change documents. The colon is followed by a capital.

AUTHORS:

corrected

REFEREE #3:

The authors perfectly tackle the recommendation made by the reviewers. This paper is in a good shape to impact the future readers.

AUTHORS:

We thank the reviewer for his/her positive evaluation of our previous revision.

REFEREE #3:

 Nonetheless, three technical problems remain:

- The first one is related to line 489 to 491 where the authors found an explanation to the lack of overlap between the disturbance events. The answer to that question can be easily set up with a few model simulations. I can understand here, that the authors can't run new simulations for this study but the authors can at least explain which simulations they will set up to disentangle the effect of wind direction and forest resilience.

AUTHORS:

The referee is right that this would be a good model exercise in subsequent studies. We substantiated our discussion following the referee's advice in l. 504-508 of the revised manuscript (track-changed version).

REFEREE #3:

- The second one is related to the last paragraph of section 4.1 (l494-504). In this paragraph, the authors conclude that their study supports their expectations of an amplifying effect of past land use on recent disturbance activity. Given the small number of cumulative NEE (-3.1tC ha-1) and the high p-value (0.191), the authors need to discuss why they don't observe a stronger effect of the natural disturbances legacy on the cumulative NEE.

AUTHORS:

The last paragraph of section 4.1 refers to the results in 3.2:

 "Based on our simulations we found only a moderate positive effect of the first disturbance episode on the volume disturbed during the second episode (+8,181 m³, p=0.401). In contrast, land use had a considerable impact on the second disturbance episode. On average, land use increased the volume disturbed by +28.927 m³ (p<0.001)."

The discussion about the role of land use and disturbances can be found in section 4.2. Here, we also discuss the reasons why the legacy effect from land use was much stronger than the one of disturbances, see e.g. l. 529-539:

"We found long-lasting legacy effects of both past natural disturbance and land use on the forest carbon cycle (see also Gough et al., 2007; Kashian et al., 2013; Landry et al., 2016; Nunery and Keeton, 2010), supporting our hypothesis regarding the importance of legacies for future C dynamics. While the legacy effect of past land use was strong, the impact of natural disturbances on the future NEE was an order of magnitude lower (Fig. 4). Here it is important to note that our results are strongly contingent on the intense and century-long land use history in Central Europe. A dynamic landscape simulation study for western North America, for instance, emphasized the dominant role of natural disturbances to determine future NEE (Loudermilk et al., 2013). In our study system, however, land use legacies may have a stronger effect on future NEE than past natural disturbances and future changes in climatic conditions (Fig. 4)."

Or l. 559-564:

"The specific disturbance history of our study area, characterized by an intensive disturbance and land use history and major socio-ecological transitions throughout the 20[th] century, is key for interpreting our findings. In particular, the cessation of forest management in 1997 had a very strong impact on the future carbon balance of the landscape (an on average 52.8 and 13.4 times higher effect than the first and second episodes of natural disturbances, respectively – see Fig. 4)."

We thus feel, we have already addressed this issue sufficiently, and extended our explanation why forest management had a stronger effect than natural disturbances on NEE only slightly in l. 539-540.

REFEREE #3:

- The third one is related to figure 4. Given the information shared by the authors in the paper and the supplementary material, I don't understand why, in figure 4, the divergence in the cumulative NEE starts in 2013 for past land use and 1st disturbance episode graphs. The authors need to explain that a least in the caption and better in the result or method section.

AUTHORS:

Our analysis refers to the legacy effects of past land use and disturbance (before 2014) and future climate (after 2013) on the future trajectories of NEE which is the reason why NEE starts with 0 in year 2013. We added an explanation in the methods section l. 404-405 as well as in the figure caption.

REFEREE #3:

Congratulations on the work done in this paper!

AUTHORS:

We are grateful for the helpful comments to improve our manuscript.

**Legacies of past land use have a stronger effect on forest carbon exchange**

**than future climate change in a temperate forest landscape**

**Running head**: "Land use legacies determine C exchange"

Dominik Thom[*, 1,2], Werner Rammer[1], Rita Garstenauer[3,4], Rupert Seidl[1]

[1] Institute of Silviculture, Department of Forest- and Soil Sciences, University of Natural

Resources and Life Sciences (BOKU) Vienna, Peter-Jordan-Straße 82, 1190 Vienna, Austria

[2] Rubenstein School of Environment and Natural Resources, University of Vermont, 308i Aiken

Center, Burlington, VT 05405, USA. Tel: +1 802 557 8221. Fax: +1 802 656 2623. Email:

dominik.thom@uvm.edu

[3] Institute of Social Ecology, Department of Economics and Social Sciences, University of

Natural Resources and Life Sciences (BOKU) Vienna, Peter-Jordan-Straße 82, 1190 Vienna,

Austria

[*] Corresponding author

**Abstract**

Forest ecosystems play an important role in the global climate system, and are thus intensively discussed in the context of climate change mitigation. Over the past decades temperate forests were a carbon (C) sink to the atmosphere. However, it remains unclear to which degree this C uptake is driven by a recovery from past land use and natural disturbances or ongoing climate change, inducing high uncertainty regarding the future temperate forest C sink. Here our objectives were (i) to investigate legacies within the natural disturbance regime by empirically analyzing two disturbance episodes affecting the same landscape 90 years apart, and (ii) to unravel the effects of past land use and natural disturbances as well as future climate on 21st century forest C uptake by means of simulation modelling. We collected historical data from archives to reconstruct the vegetation and disturbance history of a forest landscape in the Austrian Alps from 1905 to 2013. The effects of legacies and climate were disentangled by individually controlling for past land use, natural disturbances, and future scenarios of climate change in a factorial simulation study. We found only moderate spatial overlap between two episodes of wind and bark beetle disturbance affecting the landscape in the early 20th and 21st century, respectively. Our simulations revealed a high uncertainty about the relationship between the two disturbance episodes, whereas past land use clearly increased the impact of the second disturbance episode on the landscape. The future forest C sink was strongly driven by the cessation of historic land use, while climate change reduced forest C uptake. Compared to land use change the two past episodes of natural disturbance had only marginal effects on the future carbon cycle. We conclude that neglecting legacies can substantially bias assessments of future forest dynamics.

**Key words**: bark beetles, climate change, forest history, forest management, Kalkalpen National Park, legacy effects, net ecosystem exchange, wind

**Copyright statement**

The authors agree to the copyright statement as described at https://www.biogeosciences.net/about/licence_and_copyright.html.

[revised manuscript text omitted]